# Systematic examination of low-intensity ultrasound parameters on human motor cortex excitability and behavior

Anton Fomenko[1†]*, Kai-Hsiang Stanley Chen[1,2†], Jean-François Nankoo[1], James Saravanamuttu[1], Yanqiu Wang[1], Mazen El-Baba[1], Xue Xia[3], Shakthi Sanjana Seerala[4], Kullervo Hynynen[4], Andres M Lozano[1,5]*, Robert Chen[1,3]*

[1]Krembil Research Institute, University Health Network, Toronto, Canada; [2]Department of Neurology, National Taiwan University Hospital Hsin-Chu Branch, Hsin-Chu, Taiwan; [3]Division of Neurology, Department of Medicine, University of Toronto, Toronto, Canada; [4]Sunnybrook Research Institute, Toronto, Canada; [5]Division of Neurosurgery, Department of Surgery, Toronto Western Hospital, University of Toronto, Toronto, Canada

*For correspondence:
anton.fomenko@uhnresearch.ca (AF);
lozano@uhnresearch.ca (AML);
robert.chen@uhn.ca (RC)

[†]These authors contributed equally to this work

Competing interests: The authors declare that no competing interests exist.

**Abstract** Low-intensity transcranial ultrasound (TUS) can non-invasively modulate human neural activity. We investigated how different fundamental sonication parameters influence the effects of TUS on the motor cortex (M1) of 16 healthy subjects by probing cortico-cortical excitability and behavior. A low-intensity 500 kHz TUS transducer was coupled to a transcranial magnetic stimulation (TMS) coil. TMS was delivered 10 ms before the end of TUS to the left M1 hotspot of the first dorsal interosseous muscle. Varying acoustic parameters (pulse repetition frequency, duty cycle, and sonication duration) on motor-evoked potential amplitude were examined. Paired-pulse measures of cortical inhibition and facilitation, and performance on a visuomotor task was also assessed. TUS safely suppressed TMS-elicited motor cortical activity, with longer sonication durations and shorter duty cycles when delivered in a blocked paradigm. TUS increased $GABA_A$-mediated short-interval intracortical inhibition and decreased reaction time on visuomotor task but not when controlled with TUS at near-somatosensory threshold intensity.

## Introduction

There is a pressing need to develop precise and effective methods of non-invasive brain stimulation, both as tools to investigate neurophysiology and as potential therapeutic modalities for circuitopathies such as Parkinson's disease. Low-intensity transcranial ultrasound (TUS) is a promising noninvasive brain stimulation technique actively being studied for its ability to reversibly modulate mammalian brain activity (*Fomenko et al., 2018*; *Naor et al., 2016*; *Tyler et al., 2008*). By focusing the propagation of acoustic waves through the skull, a higher degree of spatial specificity and deeper targeting can be achieved over other noninvasive stimulation methods such as transcranial magnetic stimulation (TMS) and transcranial direct-current stimulation (*Fomenko and Lozano, 2019*; *Pasquinelli et al., 2019*).

Proof-of-concept studies using TUS on healthy human volunteers have elicited a broad range of suppressive and inhibitory effects on cortical and subcortical target areas, although it remains unknown how varying the sonication parameters (i.e.: sonication duration, duty cycle, frequency) affects the magnitude and direction of neural activity within human brain circuits (*Ai et al., 2018*; *Lee et al., 2016b*; *Legon et al., 2014*; *Sanguinetti et al., 2014*). To date, published human experiments have demonstrated electrophysiological suppression of sensory-evoked potentials, task

performance alteration, modulation of cortical oscillatory dynamics, and corresponding activation on fMRI (*Lee et al., 2016a*; *Legon et al., 2014*; *Mueller et al., 2014*). Self-reported outcomes induced by TUS have included mood improvement (*Hameroff et al., 2013*), phosphene detection (*Lee et al., 2016b*; *Schimek et al., 2020*), and tactile limb sensations (*Lee et al., 2015*; *Lee et al., 2016a*), when frontal, somatosensory, and occipital cortices were targeted, respectively. However, each study examined only a single set of stimulation parameters, and it is not known whether human neural circuits are more sensitive to titration of a particular parameter, nor whether a dose-response effect exists. Furthermore, studies reported heterogeneous effects of TUS on targets such as the motor cortex, with some studies showing suppression of motor-evoked potentials (MEPs) (*Legon et al., 2018a*), and others demonstrating increased cortical excitability after prolonged soni-cation (*Gibson et al., 2018*).

Here, we describe the first double-blinded study examining the effects of systematically varying sonication parameters of TUS applied to the primary motor cortex (M1) of healthy human partici-pants (*Figure 1*). We tested the M1 because measurement of MEPs from TMS provides an objective and readily quantifiable measure of M1 excitability. Using a combined TMS-TUS stimulation approach (*Legon et al., 2018a*), we hypothesize that some parameters have a linear suppressive dose-response effect, and others may be less important. We then examine the effects of TUS on sev-eral intracortical circuits using paired-pulse TMS. Finally, we study the effects of applying TUS to M1 in a visuo-motor behavioral task, to determine whether sonication alone can affect higher order synaptically connected circuits involved in motor initiation and execution.

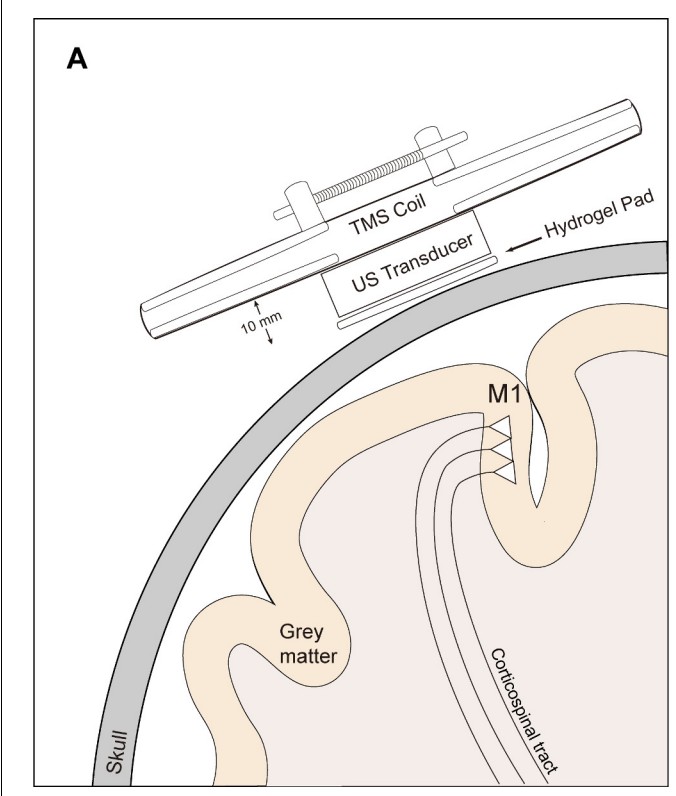
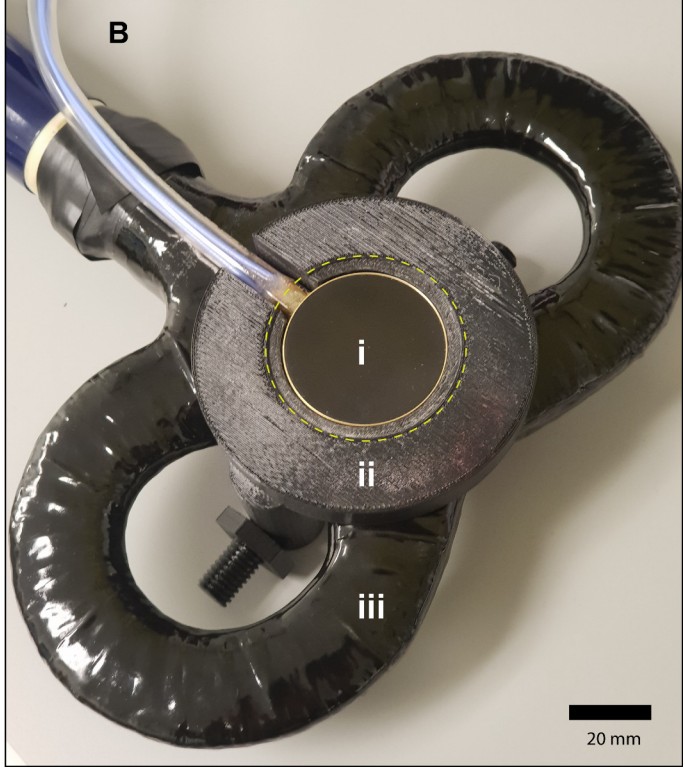

**Figure 1.** Experimental setup. (**A**) Diagram depicting the primary motor cortex hand knob in coronal section, with the ultrasound transducer coupled to the scalp via compressible hydrogel and held to the underside of a transcranial magnetic stimulation (TMS) coil with a 3D printed plastic holder (not to scale). The transducer and holder measure 10 mm thick, allowing for adequate magnetic stimulation of cortical neuron populations. (**B**) Photograph of the custom TUS-TMS delivery apparatus components, showing: (i) the active face of the TUS transducer, (ii) plastic 3D-printed holder, and (iii) 70 mm figure-eight TMS coil. The yellow dashed line indicates the recessed cutout for the hydrogel coupling pad.

The online version of this article includes the following figure supplement(s) for figure 1:

**Figure supplement 1.** Material specifications for the 500 kHz 2-element annular ultrasound transducer, and scale photographs of coupling to TMS coil.

# Results

## Safety and precision of acoustic and magnetic stimulation

At basic parameters, the estimated intracranial intensity of TUS at basic parameters was calculated to correspond to $I_{SPTA}$ = 0.69 W/cm$^2$, $I_{SPPA}$ = 2.32 W/cm$^2$, and a mechanical index of 0.19. The FDA cephalic acoustic exposure guidelines are defined as $I_{SPTA} \leq$ 94 mW/cm$^2$, and either MI $\leq$ 1.9 or derated $I_{SPPA} \leq$ 190 W/cm$^2$ (*Duck, 2007*; *United States Food and Drug Administration, 2017*). Intensity values for other parameters examined in this study can be found in *Table 1*. Cumulative verum sonication time per participant ranged between 157.7 and 192.5 s. No subjects reported any adverse effects, but two (13%) participants reported a transient warm sensation at the sonication site after multiple experimental blocks. Neurological examinations after each experimental visit were normal, supporting the safety profile of the current pulsing schemes. The probabilistic averaged map of the induced electromagnetic current across all participants (n = 16) confirmed that the M1 hand knob was effectively targeted with TMS (*Figure 2A*). Similarly, the individual simulations of ultrasound propagation for each participant confirmed acoustic targeting of a portion of M1, as well as underlying corticospinal white matter. (*Figure 2—figure supplement 1*). The focal length within the −6 dB of the focus was 23 mm, and the focal width at half-maximum of our transducer was determined to be 6 mm (*Figure 3*).

## Effects of TUS parameters on single-pulse MEPs

The mean TMS stimulator intensity required to generate a 0.5 mV MEP was 58% (range 50–80%). Basic parameters suppressed MEP amplitudes (0.33 ± 0.06 mV; n = 12) compared to active sham (0.63 ± 0.09 mV; n = 12; two-tailed paired t-test, t = 6.36, df = 11, p<0.001), and basic parameters also suppressed mean MEP voltage (0.21 ± 0.04 mV; n = 4) compared to inactive sham (0.49 ± 0.09 mV; n = 4; two-tailed paired t-test, t = 5.41, df = 3, p=0.012) (*Figure 4*).

Varying the duty cycle showed a significant effect on MEP amplitude (RM one-way ANOVA, F = 5.98, df = 15, p=0.015). Compared to sham (0.63 ± 0.14 mV; n = 16), post-hoc two-tailed paired t-tests with correction for multiple comparisons showed that TUS at a duty cycle of 10% suppressed MEPs significantly (0.36 ± 0.06 mV; n = 16; t = 2.93, df = 15, adjusted p=0.027). Sonication at 30% DC (0.48 ± 0.10 mV; n = 16; t = 2.08, df = 15, adjusted p=0.10) and 50% DC (0.44 ± 0.08 mV; n = 16; t = 2.12, df = 15, adjusted p=0.10) had no significant effect (*Figure 5*).

Varying the sonication duration also had a significant effect on MEP amplitude (RM one-way ANOVA, F = 15.12, df = 15, p<0.001). Post-hoc two-tailed paired t-tests with correction for multiple comparisons showed significant suppression compared to sham (0.89 ± 0.13 mV; n = 16) when sonicating for 0.4 s (0.47 ± 0.08 mV; n = 16; t = 4.29, df = 15, adjusted p=0.003), and 0.5 s (0.47 ± 0.11 mV; n = 16; t = 4.12, df = 15, adjusted p=0.004). Sonication for 0.1 s (0.87 ± 0.15 mV; n = 16; t = 0.28, df = 15, adjusted p=1.00), 0.2 s (0.84 ± 0.14 mV; n = 16; t = 0.73, df = 15, adjusted p=0.92), or 0.3 s (0.66 ± 0.11 mV; n = 16; t = 2.33, df = 15, adjusted p=0.12) was not significantly different from sham stimulation.

Varying the PRF with fixed DC did not have a significant effect on evoked potentials (RM one-way ANOVA, F = 2.90, p=0.08). Compared to the sham condition (0.63 ± 0.12 mV; n = 16), post-hoc two-tailed t-tests with correction for multiple comparisons did not reveal a difference for 200 Hz (0.41 ± 0.07 mV; n = 16; t = 1.96, df = 15, adjusted p=0.162), 500 Hz (0.41 ± 0.07 mV; n = 16;

**Table 1.** Calculated extracranial and estimated intracranial acoustic intensity values by parameter.

| | 10% DC, PRF = 1000 Hz, SD 0.1–0.5 s | | 30% DC, PRF = 1000 Hz, SD 0.1–0.5 s | | 50% DC, PRF = 1000 Hz, SD 0.1–0.5 s | |
| --- | --- | --- | --- | --- | --- | --- |
| | Extracranial (quantified) | Intracranial (estimated) | Extracranial (quantified) | Intracranial (estimated) | Extracranial (quantified) | Intracranial (estimated) |
| $I_{SPTA}$ (W/cm$^2$) | 0.93 | 0.23 | 2.78 | 0.69 | 4.63 | 1.16 |
| $I_{SPPA}$ (W/cm$^2$) | 9.26 | 2.32 | 9.26 | 2.32 | 9.26 | 2.32 |
| MI (unitless) | 0.74 | 0.19 | 0.74 | 0.19 | 0.74 | 0.19 |

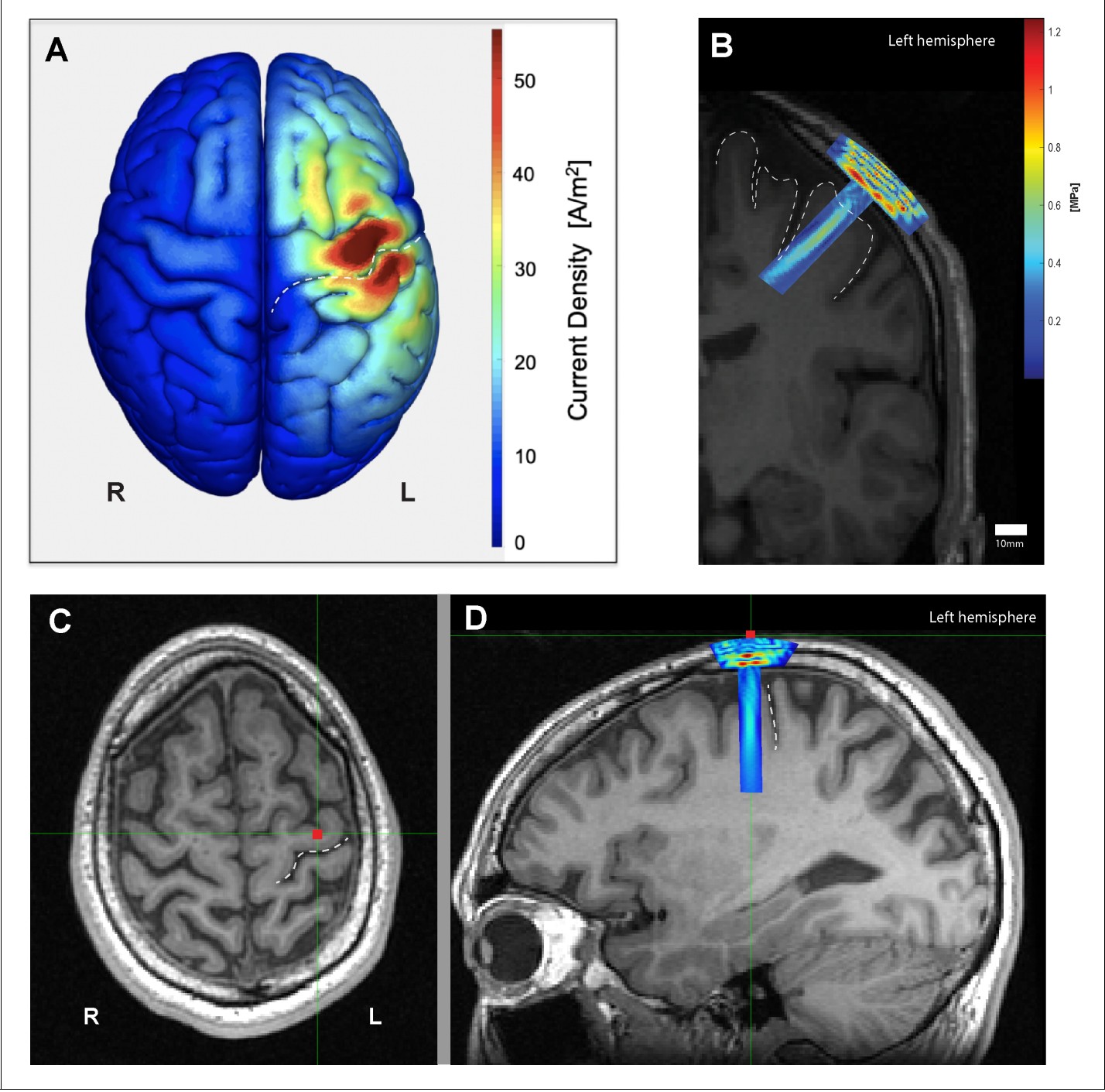

**Figure 2.** Electromagnetic and acoustic characterization. (**A**) Characterization of TMS-induced electromagnetic current distribution around the left primary motor cortex, anterior to the central sulcus (dashed line), averaged across 16 participants rendered using SIMNIBS. (**B**) Simulation of transcranial ultrasound pressure field in a characteristic participant (k-Wave MATLAB toolbox), showing high pressure at the skull, and a cigar-shaped volume of tissue activation over the motor cortex and underlying white matter, with a focus centered 30 mm away from the face. (**C**) Scalp position of the US transducer on the same participant as determined by neuronavigation software during the experiment on axial and (**D**) sagittal views, showing the underlying hand knob of the precentral gyrus, anterior to the central sulcus (dashed line).

The online version of this article includes the following figure supplement(s) for figure 2:

**Figure supplement 1.** Individual acoustic simulations of each participant's transducer focal field overlaid on the corresponding coronal T1 MRI.

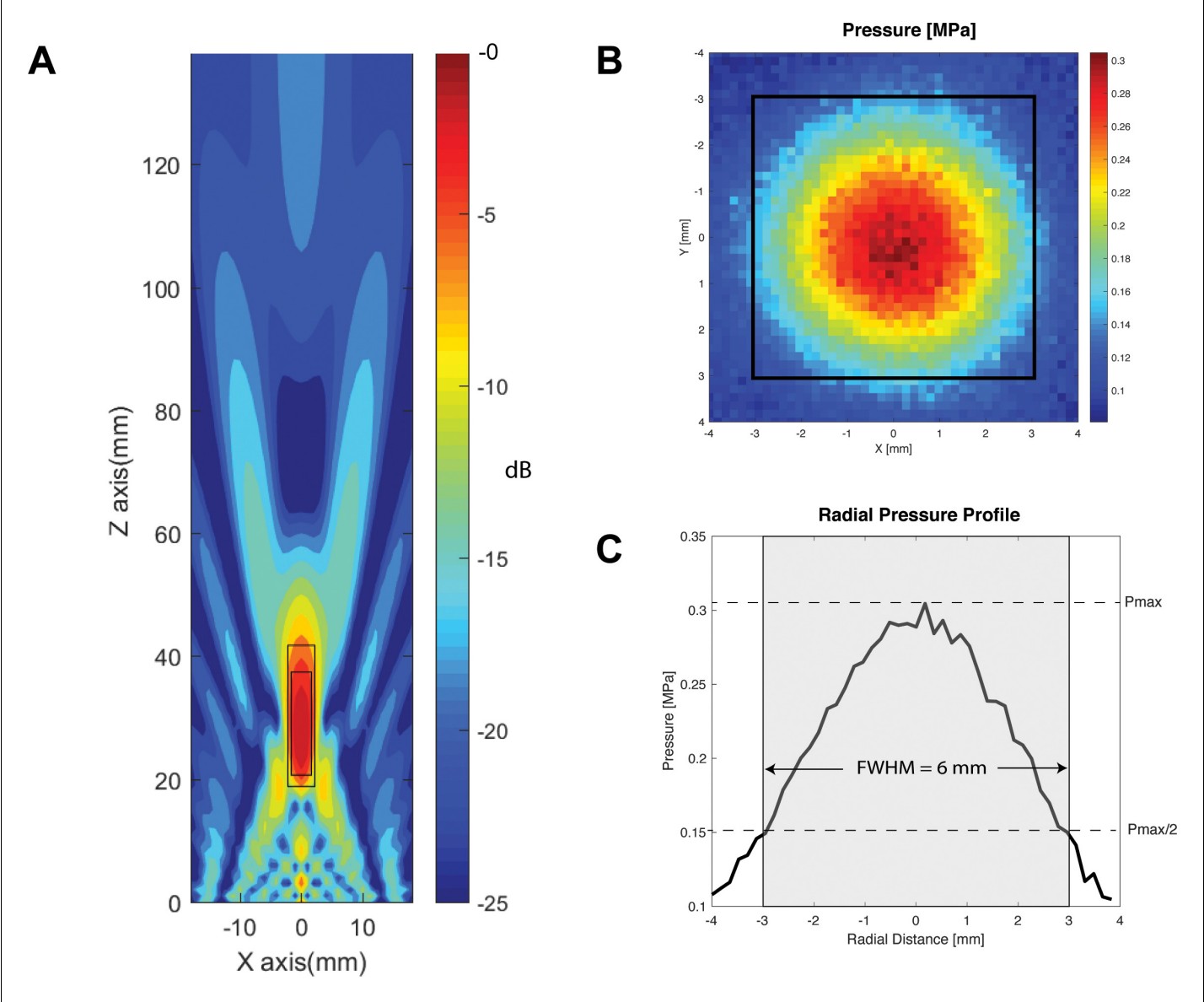

**Figure 3.** Transducer characterization. (A) Characterization of the longitudinal (Z-axis) of the two-element 500 kHz transducer with focus 30 mm away from the face, within a simulated free-water field. Within −6 dB and −3 dB of the focal point, the focal lengths are 23 mm (large rectangle), and 17 mm (small rectangle), respectively. (B) Radial (X and Y axes) hydrophone quantification of the transducer at 30 mm away from the active face (top), with focal width at half maximum outlined in black square. (C) Radial pressure profile through the origin demonstrating a focal width at half maximum of 6 mm.

t = 1.94, df = 15, adjusted p=0.170), or 1000 Hz (0.42 ± 0.08 mV; n = 16, t = 1.86, df = 15, adjusted p=0.193).

Similarly, varying the PRF with adjusted DC to ensure a fixed pulse duration did not have a significant effect on MEPs (RM one-way ANOVA, F = 1.22, p=0.31). Compared to the sham condition (0.56 ± 0.09 mV; n = 16), post-hoc two-tailed t-tests with correction for multiple comparisons did not reveal a difference for 200 Hz (0.50 ± 0.08 mV; n = 16; t = 0.92, df = 15, adjusted p=0.689), 500 Hz (0.44 ± 0.09 mV; n = 16; t = 1.67, df = 15, adjusted p=0.261), or 1000 Hz (0.55 ± 0.11 mV; n = 16, t = 0.14, df = 15, adjusted p=1.00).

## Effects of blocked versus interleaved parameter variation

Varying the duty cycle with interleaved trials (*Figure 5—figure supplement 1*) showed a significant effect on MEP amplitudes (n = 16, F = 3.94, df = 15, p=0.02; RM one-way ANOVA). Compared to

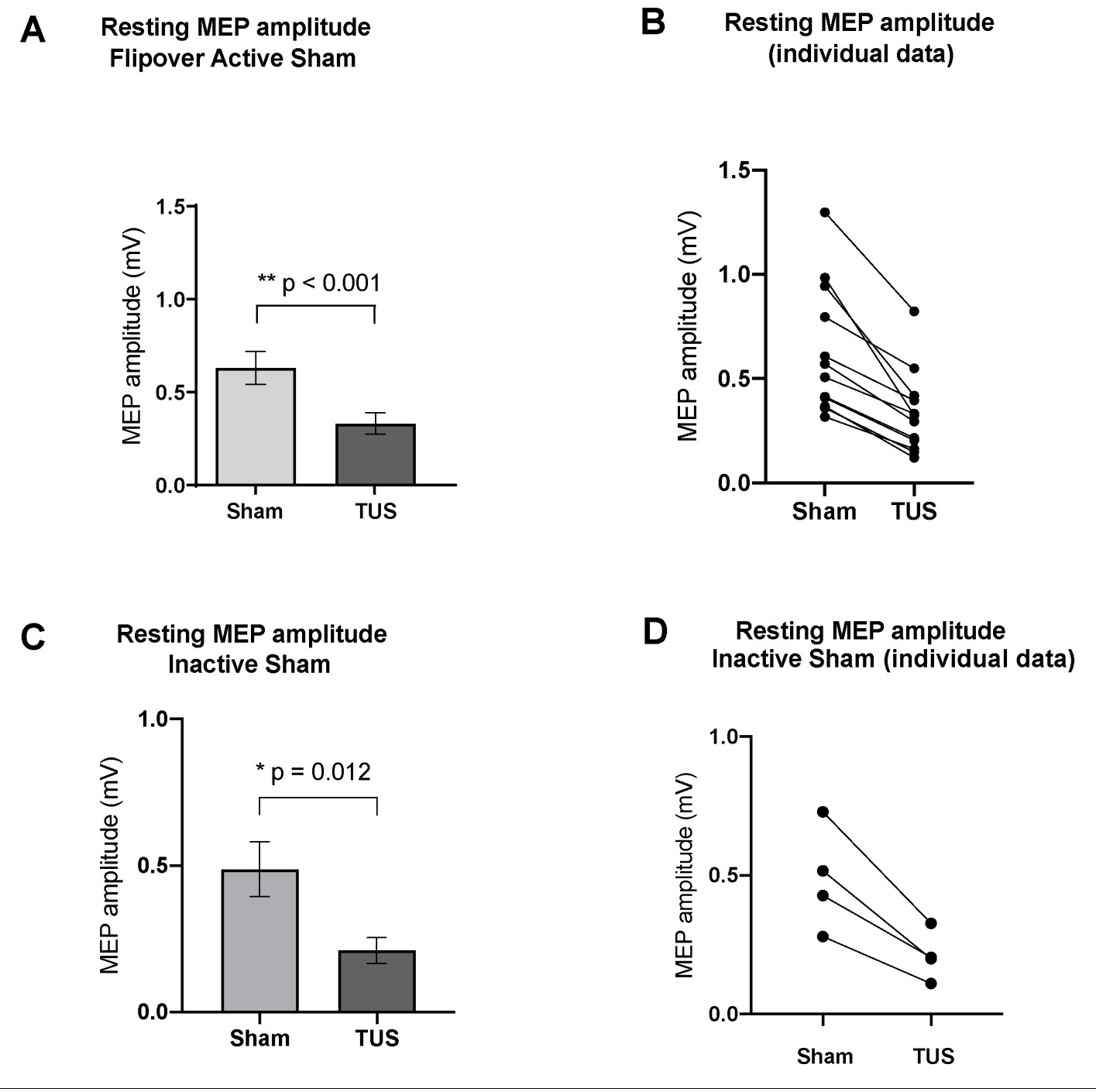

**Figure 4.** Effect of baseline ultrasound versus active and inactive sham on TMS-induced resting motor-evoked potential (MEP) amplitudes as measured by FDI EMG. (**A**) Baseline parameters (PRF = 1000 Hz, DC = 30%, SD = 0.5 s) suppressed mean MEP voltage compared to active sham, or powered transducer pointing upward (p<0.001, paired t-test) N = 12. (**B**) Individual MEP values by participant by condition (**C**) Baseline parameters suppressed mean MEP voltage compared to inactive sham, or unpowered transducer pointing toward the scalp (p=0.012, paired t-test) N = 4. (**d**) Individual MEP values by participant by condition. Error bars represent standard error.

sham (0.56 ± 0.06 mV; n = 16), post-hoc two-tailed paired t-tests with correction for multiple comparisons showed that a DC of 10% (0.41 ± 0.05 mV; n = 16; t = 3.12, df = 15, adjusted p=0.020) significantly suppressed MEPs, while a DC of 30% (0.48 ± 0.05 mV; n = 16; t = 1.47, df = 15, adjusted p=0.35) and 50% (0.45 ± 0.07 mV; n = 16; t = 2.06, df = 15, adjusted p=0.14) did not.

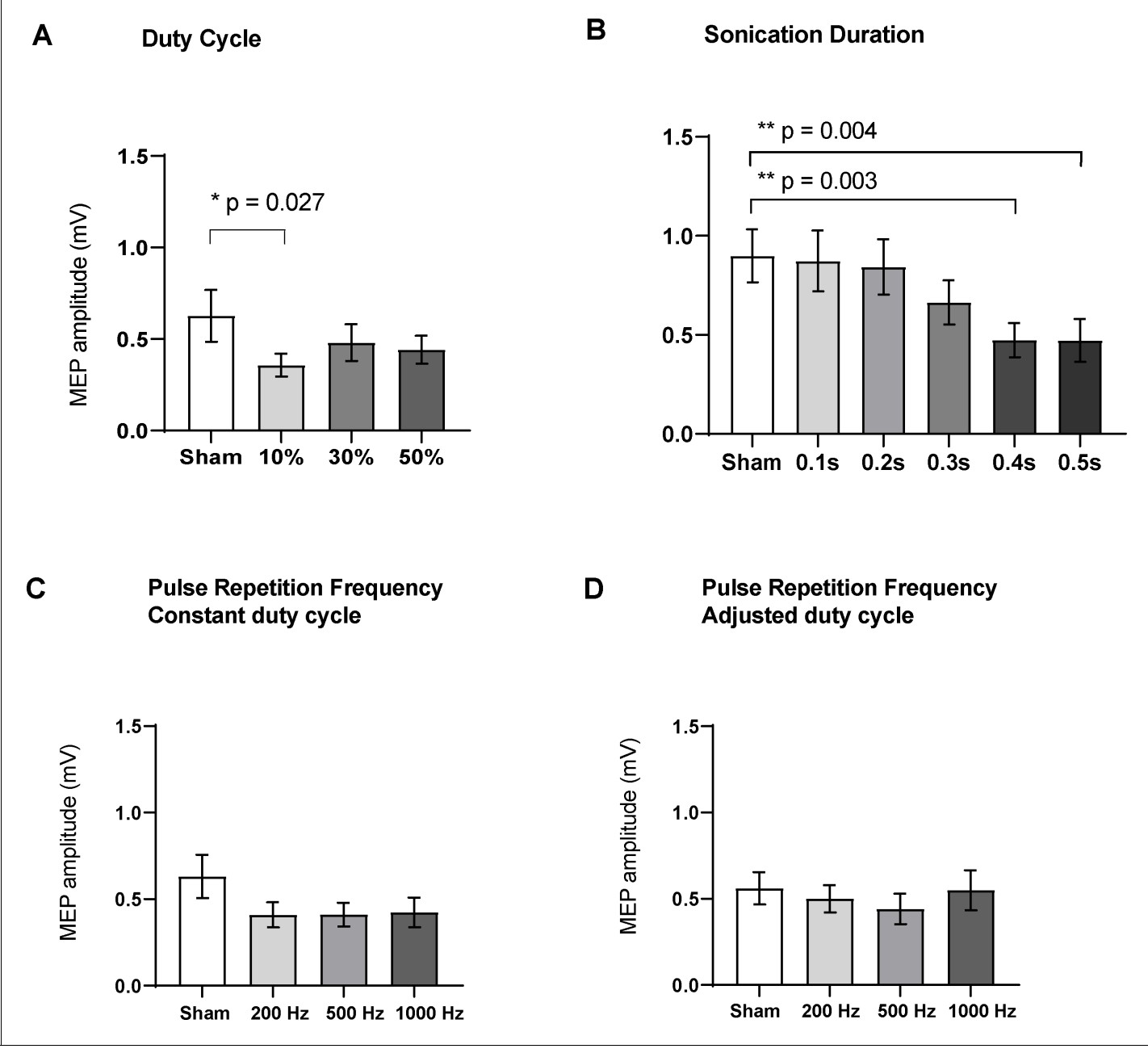

**Figure 5.** Effects of ultrasound parameters on TMS-induced resting peak-to-peak motor-evoked potential (MEP) amplitudes as measured by FDI EMG (N = 16). Means of MEPs were plotted across the different sub-experiments which varied different parameters (**A**) Duty cycle (p=0.015; RM one-way ANOVA), 10% DC suppressed MEPs compared to sham (p=0.027, paired t-test). (**B**) Sonication duration had an effect proportional to the length of sonication (p<0.001; RM one-way ANOVA), with significant suppression compared to sham with 0.4 s (p=0.003), and 0.5 s (p=0.004). (**C**) Varying the pulse repetition frequency with fixed DC (p=0.08; RM one-way ANOVA) or (**D**) adjusted DC to keep constant burst duration (p=0.31; RM one-way ANOVA) did not have a significant effect. Error bars represent standard error. Asterisks are indicative of a significant post-hoc two-tailed paired t-test, and p-values are adjusted with the Holm-Bonferroni method (α = 0.05).

The online version of this article includes the following figure supplement(s) for figure 5:

**Figure supplement 1.** Effects of blocked and interleaved variation of ultrasound parameters on TMS-induced resting peak-to-peak MEP amplitudes as measured by FDI EMG (N = 16).

**Figure supplement 2.** Post-hoc analysis of blocked delivery of ultrasound parameters by trial number (N = 16).

Blocked variation of the DC also showed a significant effect on MEP amplitudes (n = 16, F = 11.2, df = 15, p<0.0001; RM one-way ANOVA). Compared to sham (0.59 ± 0.07 mV; n = 16), post-hoc two-tailed paired t-tests with correction for multiple comparisons showed that a DC of 10% (0.34 ± 0.04 mV; n = 16; t = 3.51, df = 15, adjusted p=0.010) and 30% (0.28 ± 0.05 mV; n = 16; t = 4.54, df = 15, adjusted p=0.002) significantly suppressed MEPs, while a DC of 50% (0.49 ± 0.06 mV; n = 16; t = 1.59, df = 15, adjusted p=0.297) had no significant effect. Post-hoc analysis of individual trials (*Figure 5—figure supplement 2*) showed no significant differences between early and late trials in the sham, 10%, 30%, or 50% conditions (n = 16, adjusted p=0.31, 0.31, 0.81, 0.81; paired t-tests).

Randomly interleaving the pulse repetition frequency with constant DC parameters did not result in significant differences across conditions (n = 16, F = 2.24, df = 15, p=0.1226; RM one-way ANOVA). However, grouping parameters in a blocked paradigm resulted in significant effects on MEP amplitudes (n = 16, df = 15, F = 14.0, p<0.001; RM one-way ANOVA). Compared to sham (0.75 ± 0.10 mV; n = 16), post-hoc two-tailed paired t-tests with correction for multiple comparisons showed that a PRF of 200 Hz (0.56 ± 0.10 mV; n = 16; t = 4.43, df = 15, adjusted p=0.001), 500 Hz (0.50 ± 0.09 mV; n = 16; t = 5.29, df = 15, adjusted p<0.001), and 1000 Hz (0.37 ± 0.07 mV; n = 16; t = 5.53, df = 15, adjusted p<0.001) all significantly suppressed MEPs. Post-hoc analysis of individual trials showed no significant differences between early and late trials in the sham, 100, 500, or 1000 Hz conditions when DC was held constant (n = 16, adjusted p=0.09, 1.0, 1.0, 1.0; paired t-tests).

Interleaving the pulse repetition frequency with adjusted DC parameters did not result in significant differences across conditions (n = 16, F = 0.50, df = 15, p=0.648; RM one-way ANOVA). However, grouping parameters in a blocked paradigm resulted in significant effects on MEP amplitudes (n = 16, df = 15, F = 9.39, p=0.003; RM one-way ANOVA). Compared to sham (0.72 ± 0.11 mV; n = 16), post-hoc two-tailed paired t-tests with correction for multiple comparisons showed that a PRF of 200 Hz (0.50 ± 0.09 mV; n = 16; t = 3.49, df = 15, adjusted p=0.011), 500 Hz (0.45 ± 0.09 mV; n = 16; t = 5.33, df = 15, adjusted p<0.001), and 1000 Hz (0.42 ± 0.09 mV; n = 16; t = 3.16, df = 15, adjusted p=0.021) all significantly suppressed MEPs. Post-hoc analysis of individual trials showed no significant differences between early and late trials in the sham, 100, 500, or 1000 Hz conditions (n = 16, adjusted p=0.92, 1.0, 1.0, 1.0; paired t-tests).

## Effects of TUS on active MEP

During muscle contraction, active MEP amplitudes did not significantly differ between sham (7.65 ± 0.75 mV; n = 12) compared to TUS (7.79 ± 0.88 mV; n = 12; two-tailed paired t-test, t = 0.44, df = 11, p=0.67). The silent period (SP) duration after sham (131 ± 10 ms; n = 12) did not significantly differ from TUS (138 ± 9 ms; n = 12; two-tailed paired t-test, t = 0.52, df = 11, p=0.61). (*Figure 6*).

## Paired pulse TMS-TUS

Among all subjects, the mean TMS stimulator output required to produce a 1 mV MEP with sonication at basic parameters (78.8, range 55–97% MSO; n = 12) was significantly higher (mean of differences = +1.3%; n = 12; two-tailed paired t-test, t = 2.86, df = 11, p=0.02) than the output required to produce a 1 mV MEP under sham (77.4 range 57–99% MSO; n = 12) (*Figure 7*).

Short-interval intracortical *i*inhibition at 2 ms was significantly increased with TUS application at basic parameters (0.40 ± 0.07 ratio to TS; n = 12) compared to sham (0.60 ± 0.10 ratio to TS; n = 12; two-tailed paired t-test, t = 2.32, df = 11, p=0.038). Long-interval intracortical inhibition (LICI) at 100 ms, short-interval intracortical facilitation (SICF) at 1.5 ms, SICF at 2.9, and intracortical facilitation (ICF) at 10 ms revealed no significant differences between sham and active TUS application time-locked to the first stimulus (n = 12; two-tailed t-tests, df = 11, p-values 0.91, 0.66, 0.75, 0.90, respectively).

## Behavioral task

The pooled reaction time to the presentation of visual stimulus was significantly shorter with TUS applied at basic intensity and parameters (363 ± 68 ms; n = 12) compared to sham stimulation (521.57 ± 20 ms; n = 12; two-tailed paired t-test, t = 2.27, df = 11, p=0.04) (*Figure 8*).

In four additional subjects, the reaction times were not significantly different between TUS applied at 0.54 W/cm$^2$ subthreshold intensity (305 ± 45 ms; n = 4) compared to basic 2.32 W/cm$^2$

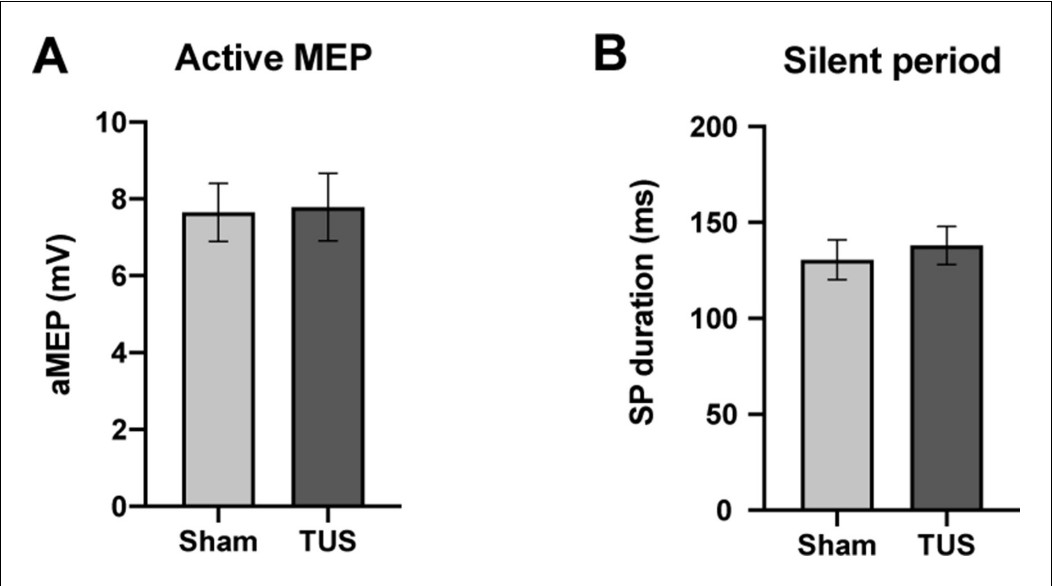

**Figure 6.** Results of single-pulse experiments investigating active MEP, including (**A**) amplitude, and (**B**) silent period (SP) duration of sham and TUS condition. N = 12. Error bars represent standard error.

intensity (301 ± 44 ms; n = 4; two-tailed paired t-test, t = 2.68, df = 3, p=0.08). There was no significant difference between application of TUS or sham in the mean total distance travelled, deviation scores, or time taken per trial (one-way, RM one-way ANOVA, p>0.05). This remained the case even when sorting by target location.

## Discussion

To our knowledge, our experiment represents the first systematic investigation of low-intensity ultrasound parameters on human cortical activity. We found a dose-response effect of prolonging the sonication duration on the suppression of TMS-induced MEPs, up to a maximum duration of 0.5 s, and show that a lower duty cycle (10%) has the greatest efficacy in suppressing motor cortex potentials. Similar studies with large animals appear to be in line with our data. A recent systematic examination of TUS parameters in sheep showed that low duty cycles (<10%) and long sonication durations (~1 min) yielded suppressive effects on sensory-evoked potentials, while high duty cycles (>30%) and short sonication durations (<500 ms) were excitatory on the motor cortex (*Yoon et al., 2019*) However, unlike our study, suppressive parameters were only applied to the sensory cortex, and the excitatory parameters to the motor cortex, and the animals were de anesthetized during the experiment.

Interestingly, our initial results show that although a particular combination of parameters can robustly suppress TMS-elicited cortical activity when applied in a sham-controlled block design (*Figure 4*), some parameters such as PRF when randomized and varied individually do not have the same robust effect, whereas others such as sonication duration do (*Figure 5C–D*). To address this discrepancy, we conducted follow-up experiments (*Figure 5—figure supplement 1*) in which we delivered three distinct parameter sets in blocked, and interleaved fashion. We found that sonication at 10% DC consistently results in suppression regardless of experimental design, whereas 30% DC only results in suppression when applied in blocked fashion. A duty cycle of 50% did not show any difference compared to sham regardless of experimental design. In contrast, we observed that sonication at three different pulse repetition frequencies (200, 500, and 1000 Hz) applied in a blocked design resulted in reduced cortical excitability compared to sham, whereas interleaving these parameters yielded no significant difference. These results held whether duty cycle was fixed, or whether the DC was adjusted to maintain an equal burst duration.

In summary, our findings suggest that when delivered in blocked fashion, longer sonication durations, lower duty cycles, and all three PRFs tested yield effective suppression of TMS-elicited MEPs.

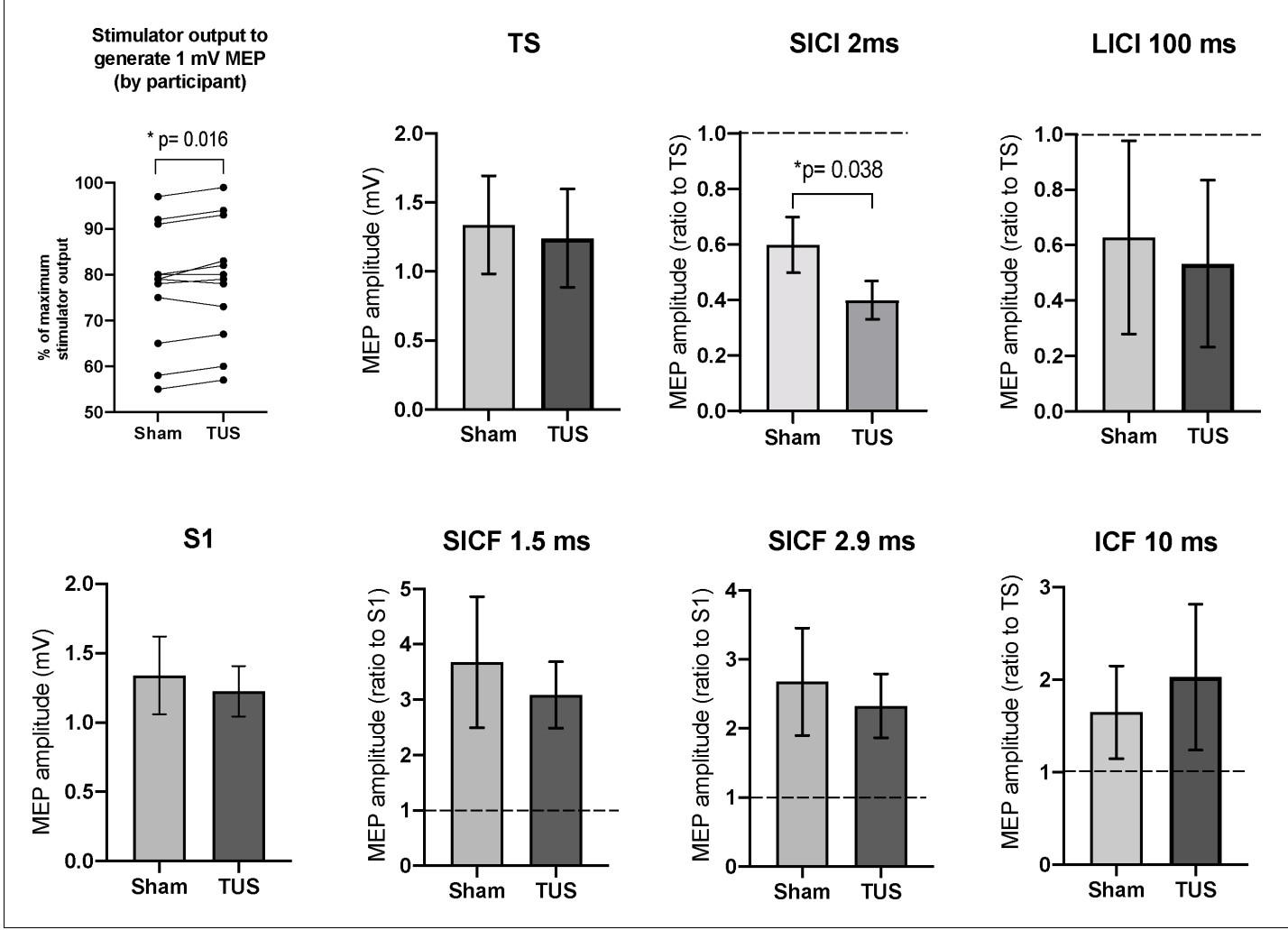

**Figure 7.** Results of the paired-pulse experiments. A mean increase of +1.3% in stimulator intensity is required to elicit a 1 mV MEP under the TUS condition, compared to sham (p=0.016, paired t-test). After adjusting stimulator output to compensate, suppression, the TS and S1 conditions are not significantly different between sham and TUS. Results shown include short-interval intracortical inhibition (SICI), intracortical facilitation (ICF), long-interval intracortical inhibition (LICI), and short-interval cortical facilitation (SICF). N = 12. The data is plotted as a ratio to the test stimulus (TS) or stimulus alone (S1) amplitude. Ratios higher than 1.0 indicate facilitation and ratios below 1.0 indicate inhibition. TUS applied before the test stimulus significantly reduced SICI compared to sham (p=0.038, paired t-test). Error bars represent standard error.

The trend toward greater suppression with increasing blocked PRF agree with recent literature, where higher PRF (1500 Hz) in combination with low duty cycles were found to be more effective than lower PRF (300 Hz) in neuromodulation of mouse motor cortex in vivo (*King et al., 2013*) and in vitro (*Manuel et al., 2020*). Furthermore, recent large animal studies reveal a bidirectional neuro-modulation effects of varying TUS parameters (*Yoon et al., 2019*). As such, interleaving different parameters in short succession may lead to spillover effects, due to the random order of parameter delivery. From animal studies where randomized delivery of TUS parameters was studied, non-linear effects were found, possibly related to non-linear piezoelectric accumulation across the neural membrane capacity under the Neuronal Bilayer Sonophore model (*Kim et al., 2014*; *Plaksin et al., 2014*). In addition, excitatory or inhibitory changes in short-term plasticity may occur with repeated TUS stimulation, similar to those observed with repeated magnetic (*Watanabe et al., 2014*) and electrical stimulation (*Udupa et al., 2016*). These are currently under study in our laboratory and in emerging reports on LITUS short-term plasticity in animal models (*Yu et al., 2019*).

To test whether a progressive accumulation of inhibition might be responsible for the observed difference in results between the blocked and interleaved study designs, we performed a post-hoc

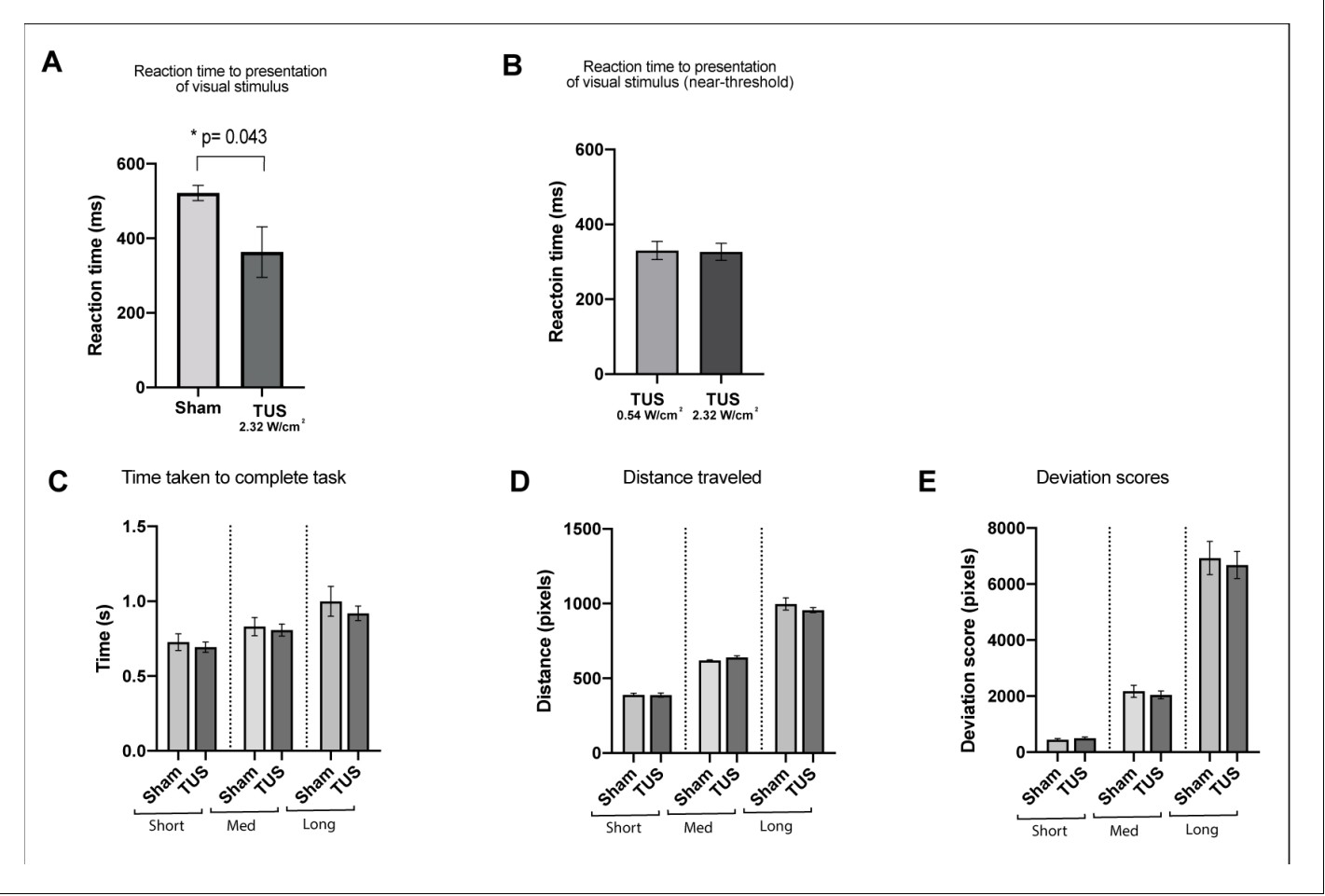

**Figure 8.** Results of the visuo-motor behavioral task experiment, showing reduction in a pooled reaction time to the presentation of visual stimulus with TUS at 2.32 W/cm$^2$, p=0.043, paired t-test (N = 12). Reaction times for TUS at 2.32 W/cm$^2$ compared to 0.54 W/cm$^2$ were not significantly different (N = 4). Bottom row: Comparison of Sham and TUS trials sorted by distance to target (near, medium, far). N = 12. Asterisk denotes significance p<0.05 on paired t-test. Error bars represent standard error.

The online version of this article includes the following figure supplement(s) for figure 8:

**Figure supplement 1.** Visuomotor behavioral task starting position and targets, as they appear on the screen during the experiment (top), and instructions given to participant (bottom).

analysis of our blocked experiments, stratifying by trial number across all participants (*Figure 5—figure supplement 2*). Within each parameter condition, we did not detect a significant difference between the magnitude of early and late trials and conclude that there is no temporal accumulation of MEP-suppressive effects over blocks of 15 trials, corresponding to a block length of approximately 90 s. Instead, the suppression appears to be instantaneous, with effective parameters increasing the likelihood of generating a lower TMS-elicited MEP, but not potentiating the effect of a subsequent stimulation. Notably, our blocked design features a 20 s pause between each set of 15 replicate stimulations, whereas interleaved delivery of random parameters involves an uninterrupted session of 60 stimulations. We speculate that this 20-s rest period may lead to a resetting of cortical excitability to a more TUS-sensitive state. The lack of resetting could play a role in our observation of less robust effects when certain parameters are randomized in succession.

Possibly because of these longer term effects and the heterogeneity of experimental methods and sonication parameters used between human studies, conflicting reports have emerged. For instance, *Legon et al., 2018a* applied simultaneous 500 kHz TUS and TMS to the M1 of healthy human subjects to examine effects on the MEP, intracortical excitability, and reaction time (*Legon et al., 2018b*). TUS was found to attenuate TMS-evoked MEP amplitude, reduce ICF, but

with no effect on intracortical inhibition, and improve reaction time. However, another study by Gibson found that application of continuous sonication for 2 min to M1 increased subsequent TMS-induced corticospinal excitability compared to baseline (*Gibson et al., 2018*). Notably, both studies were single-blinded, used a single set of sonication parameters and used a blocked study design to examine sham and active TUS conditions. The study by Gibson also used an unfocused imaging transducer operating at high fundamental frequencies (2.32 MHz), decreasing the likelihood that the acoustic energy reached the brain due to the lower transmission coefficient of the skull at this frequency range (*Mueller et al., 2017*). Since Gibson's study measured motor cortical excitability only *after* ultrasound was applied for several minutes, the potential neuromodulatory mechanisms may have opposing effects on excitability compared to the on-line TUS-TMS paradigm studied herein, and in Legon's experiments (*Gibson et al., 2018*; *Legon et al., 2018b*).

Since the piezoelectric element within our ultrasound transducer emits a slightly audible buzzing tone when activated, also reported in other human TUS-TMS studies (*Legon et al., 2018b*), we implemented a novel method of delivering a masking audio stimulus to reduce potential auditory confounding effects. This is particularly salient because our sham conditions were randomly interleaved with active TUS delivery in order to reduce the effects of time-related fluctuations of cortical excitability and to blind the experimenter (*Huber et al., 2013*). Recent literature demonstrating potential auditory confounding in rodent TUS cortical neuromodulation (*Guo et al., 2018*; *Niu et al., 2018*; *Sato et al., 2018*), highlight the importance of strategies aimed at prevent conscious or unconscious auditory contamination in ultrasound neuromodulation experiments. Human TMS studies have shown that pairing TMS with an audible stimulus, especially human speech, can lead to increases in cortical excitability. For instance, TMS to the M1 hand area, concomitant with different auditory stimuli *Flöel et al., 2003* have shown that pure tonal sounds do not alter the magnitude of MEP, while language perception significantly *increases* motor excitability. In agreement, studies examining TMS motor response of the leg (*Liuzzi et al., 2008*) and tongue (*Fadiga et al., 2002*) show a motor facilitation with semantic speech stimuli, but not with audible tonal sounds or noise. In our study, we did not find facilitation, but rather *decreased* excitability with application of TUS to the hand area of M1, while controlling for audible tones in all delivered conditions including sham. Furthermore, our audible masking tone was played for the same 0.5 s duration for every experimental condition, including the sham, relative to the TMS pulse, and we therefore expect that any TMS-acoustic pairing would be controlled for.

A recent report examining masked and unmasked delivery of TUS conditions highlights the electrophysiological basis of potential auditory confounds in human single-transducer TUS experiments (*Braun et al., 2020*). In particular, this report shows that an audio-masked TUS sonication and audio-masked sham evoke *identical* auditory ERP's, whereas TUS alone without a mask evokes a significantly different ERP, and moreover can also be reliably differentiated from sham by participants (*Braun et al., 2020*). This report appears to corroborate our strategy and highlights the importance of controlling for auditory confounds in future human ultrasound neuromodulation studies.

The results from our paired-pulse experiments suggest that TUS potentiates short-interval intracortical inhibition (SICI). Previous pharmacological and physiological investigations have shown that SICI is a marker of GABA$_A$-receptor-mediated inhibition and may reflect activity of inhibitory interneurons (*Stagg et al., 2011*; *Udupa et al., 2010*). Our finding may suggest that cortical interneurons in layers II/III which are well-encompassed within the acoustic focus (*Figure 2B*) may be preferentially sensitive to ultrasound applied for long durations and a short duty cycle and confer an overall suppressive effect by increasing GABA$_A$ activity. Indeed, emerging electrophysiology work in animal models is suggesting that sensitivity to low-intensity TUS is mediated not only by parameter selection but also that excitatory and inhibitory neurons have different sensitivities to sonication (*Yu et al., 2019*). One possible mechanobiological explanation for the cell-type selective effects of ultrasound is the neuronal intramembrane cavitation excitation (NICE) model. Predictive modeling studies within the NICE framework, corroborated with animal electrophysiological studies are suggesting that inhibitory cortical neurons are hypersensitive to discontinuous pulsed bursts of TUS (*Plaksin et al., 2014*; *Plaksin et al., 2016*). The T-type voltage-gated calcium channels within these neurons predispose to accumulation of electrical charge between short bursts of ultrasound, predisposing to inhibitory neural activation during low duty-cycle sonication (*Plaksin et al., 2016*; *Plaksin et al., 2014*).

In Legon's study, TUS was found to attenuate ICF but did not affect intracortical inhibition (*Legon et al., 2018b*). Several methodological differences from our study should be noted, such as time-locking the ultrasound to begin 100 ms prior to the conditioning stimulus, whereas our sonication was time-locked to 490 ms prior to the first TMS pulse. Since sonication duration is a key parameter of TMS-mediated EMG suppression according to our findings, the earlier onset of TUS in our experiment relative to the CS and TS pulses might have potentiated greater GABA$_A$-mediated inhibition, despite the total sonication time being the same. In addition, we randomized the TS-alone condition with the paired pulse conditions in our experiment, whereas (*Legon et al., 2018b*) compared to a baseline MEP conducted in a temporally discrete block, potentially introducing variability inherent in MEP fluctuations over time (*Ellaway et al., 1998*).

The cortical silent period, thought to be a marker of GABA$_B$ activity (*Tremblay et al., 2013*; *Ziemann et al., 2015*), did not significantly change with application of TUS in our experiments. Likewise, we did not observe an effect of TUS on LICI, another TMS marker of GABA$_B$-ergic inhibition (*Premoli et al., 2014*). Reports using voltage-clamp and in-vivo experiments are beginning to show that TUS has preferential effects on certain ion channel currents (*Kubanek et al., 2016*; *Yu et al., 2019*), and our results may suggest that the ligand-gated ionotropic GABA$_A$ channel may be such a mechanosensitive substrate.

Similar to prior reports of M1 TUS in humans (*Gibson et al., 2018*; *Legon et al., 2018b*), we were unable to evoke motor potentials or EMG activities by application of TUS alone to the motor cortex. Interestingly, studies in small rodents consistently show EMG activity and frank motor contractions of hindlimbs (*Gulick et al., 2017*), whiskers (*Mehić et al., 2014*), and tail (*Yoo et al., 2013*) contralateral to the sonicated motor cortex, as well as inducing eye movements and pupillary dilatation (*Kamimura et al., 2016*). Although the majority of these studies were done under complete or partial anesthesia, recent studies with awake, freely moving rats have also found that TUS alone can also trigger truncal and limb movements (*Lee et al., 2018*). Applying a similar paradigm to larger animals such as sheep failed to trigger visible motor contractions with TUS alone, although time-locked and parameter-dependent EMG activity in the corresponding hindlimbs was seen (*Lee et al., 2016c*; *Yoon et al., 2019*). In the even larger human brain, we speculate that the narrow acoustic focus of a single transducer targets a small proportion of the entire motor cortex and either recruits insufficient cortical neurons to trigger a contraction, or preferentially recruits an inhibitory ensemble of interneurons thereby having the opposite effect. Moreover, recent studies with deafened rodents (*Guo et al., 2018*; *Sato et al., 2018*) have challenged the notion that low-intensity TUS can directly mediate small animal motor cortical activation, instead proposing that an indirect auditory mechanism, or a peripherally mediated startle response due may be involved. The narrow ellipsoid-shaped acoustic focus may also explain why we observed an increase in the stimulator output required to generate a 1 mV MEP (*Figure 7A*) and reduction in resting MEP amplitude, but found no change in the active motor threshold or active MEP amplitude (*Figure 6*), given the active condition is likely associated with larger area of activation in the motor cortex.

## Safety

The healthy participants in this study reported no adverse effects after participating in study visits. Detailed neurological examinations after each visit were normal. The 2017 Federal Drug Administration (FDA) guidelines for adult cephalic ultrasound suggest a maximum $I_{SPTA}$ of 94 mW/cm$^2$, a maximum $I_{SPPA}$ of 190 W/cm$^2$, and a MI of <1.9 to avoid cavitation and heating (*United States Food and Drug Administration, 2017*). Prior experimental estimates of the attenuation of human skull bone show a −12 db (*Deffieux and Konofagou, 2010*) or a 3.7- to 4.1-fold drop in intensity (*Legon et al., 2014*) at the fronto-parietal bone. Based on these attenuation values and our quantification of the transducer in free water, we estimate that our intracranial intensity at the acoustic focus under baseline parameters is $I_{SPTA}$ = 0.64 W/cm$^2$, a $I_{SPPA}$ = 2.32 W/cm$^2$ and a MI of 0.19 (*Table 1*). Despite the $I_{SPTA}$ falling above the FDA acoustic exposure guidelines, in Europe, the International Electrotechnical Commission 60601-2-5 standard for low-intensity therapeutic ultrasound equipment suggests a higher limit of 3 W/cm$^2$ on effective acoustic intensity (*Duck, 2007*). Indeed, computational models have suggested that typical low-intensity TUS parameters delivered for 0.5 s durations lead to negligible increases in brain temperature (4.27 $\times$ 10$^{-3}$ °C). (*Mueller et al., 2016*). Although no human studies have reported any permanent adverse effects from application of low-intensity TUS (*Legon et al., 2020*), a single sheep study showed small microhemorrhages at the cortical site

of sonication, albeit after prolonged repetitive (>500) trials at intensities ($I_{SPPA}$ = 10.5 W/cm$^2$) higher than used in our study (*Gaur et al., 2020*; *Lee et al., 2016c*). The absence of edema, necrosis, or local inflammatory responses, as well as a lack of control group may implicate postmortem brain extraction as the cause of these findings, rather than sonication itself. Indeed, a follow-up sheep study by the same group using fewer consecutive sonications (<80 trials) and an inter-stimulus interval of 5 s, as in our study, did not yield any abnormalities on histological examination of sonicated brain tissue (*Yoon et al., 2019*).

## Behavior

We report a reduction in mean motor reaction time with M1 TUS alone applied at basic parameters and intensity compared to sham. This result is consistent with a previous study, where subjects pressed a button when a visual stimulus appeared on the screen (*Legon et al., 2018b*). Contralateral M1 sonication 100 ms prior to the stimulus resulted in a significantly shortened response time compared to sham, although the magnitude of the difference was only about 10 ms. In our study, sonication was applied 250 ms prior, and the task was more complex; nevertheless, the sonication parameters and cortical location were similar, and we observed a larger effect size , although with higher variability from subject-to-subject. A potential mechanisms for this effect is TUS-mediated enhancement of surround inhibition (*Beck and Hallett, 2011*). In animal studies, GABA-antagonist drugs applied to M1 cause merging of motor hotspots (*Schneider et al., 2002*), and in primates, corticospinal pyramidal neurons are known to activate several nearby muscles (*Andersen et al., 1975*). If M1 TUS has a direct effect on reaction time, a possible mechanism might be potentiation of GABA$_A$ inhibitory interneuron activity surrounding the FDI hotspot, with a resulting sharpening of corticospinal motor output to the target muscle effectuating the movement. Although the slightly audible sound of the piezoelectric transducer element was masked with an audible tone, we cannot rule out intersensory facilitation (*Diederich and Colonius, 1987*; *Forster et al., 2002*) as contributing to the difference in reaction times between conditions. Given that acoustic waves are mechanical in nature, it is possible that a subtle tactile sensation on the scalp at the site of transducer placement might act as an extra sensory cue which is difficult to replicate with a sham protocol that delivers no acoustic energy. In a follow-up control experiment with four participants, the reaction times were not significantly different when a lower near-somatosensory threshold intensity (0.54 W/cm$^2$) was used when compared to the 2.32 W/cm$^2$ intensity used in all other experiments. Since individual somatosensory thresholds differ, and the threshold of transcranial acoustic energy necessary to affect motor tasks is unknown, further experiments are necessary to disentangle the role of somatosensory confounding in this behavioral task.

We were unable to detect any significant difference in mean trajectory distance, deviation score, or in time taken for each trial. This is consistent with our hypothesis that low-intensity TUS has a mainly in short-term and focal effect on neural circuits. Since sonication was only maintained for a total of 250 ms from the onset of the visual cue, we did not expect to see differences between conditions in the coordination of the relatively long motor task that underlies the full trajectory.

## Limitations

From a technical standpoint, precise targeting of intracranial structures remains challenging with single-element US transducers. Although we did not use on-line neuronavigation, we estimate that the epicenter of the transducer fell directly over the primary motor cortex based on post-hoc subject-specific MR image registration and acoustic simulations (*Figure 2—figure supplement 1*), inter-participant variability in skull thickness and cortical geometry may distort the geometry and result in a diminished actual intensity at the acoustic focus (*McDannold et al., 2004*; *Mueller et al., 2017*). Because of the complex propagation of ultrasound through intervening tissues of different acoustic impedances (i.e.: gel pad, hair, trapped air bubbles, scalp, skull, dura), the precise proportion of M1 targeted by the acoustic field, nor the precise intensity at the focus cannot be precisely ascertained. In addition, simultaneous sonication of all cortical sublayers with different neural population, as well as underlying white matter may have contributed to the MEP variability we saw in some subjects. Although our acoustic simulations were based on individualized brain imaging and transducer positions were captured by neuronavigation, the simulations are limited to two-dimensions, and the tissues are treated as homogenous layers due to absence of CT-derived density data which limits the

fidelity of the estimated focus. In the behavioural task, we did not conduct formal somatosensory threshold testing for individual participants, so intersensory facilitation cannot be ruled out as responsible for the shortened reaction times. Lastly, we did not explicitly characterize the effect of the TMS-induced electromagnetic field on the operation of the ultrasound transducer, nor the effects of the transducer housing on the coil's induced electrical field. However, such a characterization was rigorously performed in a prior study (*Legon et al., 2018b*) using the same TMS coil and a similarly sized custom non-ferromagnetic ultrasound transducer, and found no significant effects in either direction.

## Conclusions

Low-intensity TUS stimulation holds appeal for its unique combination of non-invasiveness, safety, high precision, and broad range of neuromodulatory effects. We found that lower duty cycle and longer sonication duration resulted in inhibition of cortical excitability. PRF showed suppression at all three frequencies tested (200, 500, 1000 Hz), but only when delivered in blocked design. In addition, TUS increased SICI, suggesting a possible GABAergic mechanism. Although we have shown that our human M1 TUS sonication scheme is safe and produces suppressive effects, a wider range of parameters and brain regions remain to be systematically tested, and further prospective studies are needed to elucidate region- and neuron-specific sensitivity to focused ultrasound. In addition, many cortical and subcortical cerebral targets have yet to be explored in terms of potential electrophysiological and behavioral responses to ultrasound. Dedicated study designs will be needed to answer the question of both short-term and long-term effects of TUS, as well as its on-line and off-line interaction with TMS. Finally, basic in vitro electrophysiologic experiments will help illuminate potential cellular, synaptic, and ionic mechanisms of ultrasound neuromodulation. Clinical applications are already being foreshadowed with human trials underway using low-intensity sonication for the treatment of epilepsy (*Bystritsky et al., 2015*; *Stern, 2014*), Alzheimer's disease (*Meng et al., 2017*), Parkinson's disease (*Wagner and Fregni, 2012*), disorders of consciousness (*Monti et al., 2016*), and stroke (*Tsivgoulis and Alexandrov, 2007*). The current landscape of ultrasound neuromodulation holds great promise as a functional brain-mapping tool and a potential intervention against disabling brain disorders.

## Materials and methods

### Subject recruitment and study visits

Eighteen healthy participants aged 29–59 years (8 males, 10 females; 1 left-handed, 17 right-handed) were recruited from advertisements posted at the University Health Network (UHN). No participants had any known medical conditions or taking any prescription medications, and all had normal neurological examination before the experiment. All patients gave written informed consent, and the protocol was approved by the UHN Research Ethics Board in accordance with the Declaration of Helsinki on the use of human subjects in experiments.

Subjects were studied on four separate visits. The first visit involved a three-dimensional T1-weighted 3T MRI scan of the brain for all subjects, in order to later confirm the stimulated cortical target with individualized neuronavigation (Brainsight, Rogue Research, Montreal, Quebec). MRI acquisition parameters consisted of: 3-D fast spoiled gradient recalled sequence, TR = 7.8 ms, TE = Min, FA = 12, Inversion time = 450 ms, matrix = 256 × 256, FOV = 240 mm, slice thickness = 1.1 mm, no gap, number of slices = 172, acquisition time = 10 min. On the same day, a TMS threshold assessment with a 70 mm figure-eight coil held against the scalp at the first dorsal interosseous (FDI) hotspot was performed. Since a scalp-coil distance of 10 mm is used in all subsequent experiments, two participants requiring very high stimulation intensities (>95% of maximum stimulator output to generate a 1 mV peak-to-peak MEP in the contralateral FDI) were excluded, leaving 16 participants on which to perform data collection. The second visit involved delivery of single-pulse TMS-TUS in blocked design at basic parameters, followed by different sonication parameters in interleaved design, with recording of resting and active motor potentials. The third visit involved paired-pulse TMS-TUS experiments as well as a behavioral task involving application of TUS alone. The fourth visit involved varying selected sonication parameters in blocked, followed by interleaved fashion. For each acoustic parameter of interest (i.e. DC), a block was defined as 15 consecutive sonications of

the same parameter value (i.e.: DC of 50%) with an ISI of 5 s. Four blocks were administered in random order (i.e.: DC 50%, Sham, 30%, 10%), with a 20 s rest period between blocks (*Lee et al., 2016c*). In the interleaved design, parameter values are interleaved by the MATLAB delivery script for a total of 60 randomized sonications balanced across the four parameter values, with an ISI of 5 s and no rest periods. Subjects received a detailed sensorimotor neurological assessment by a neurologist before and after each study visit.

## Ultrasound transducer

A custom two-element annular array ultrasound transducer (Sonic Concepts Inc, Bothell, Washington) operating at a fundamental frequency of 500 kHz and housed in a MRI-safe non-ferromagnetic brass cylinder measuring 38 mm in diameter and 10 mm thick was used (*Figure 1—figure supplement 1*). The transducer was coupled to a programmable two-channel 0–80 Watt radiofrequency amplifier (Sonic Concepts Inc, Bothell, Washington) via a 50 Ω impedance matching module. The amplifier contained an Arduino module controlling the phasing of a two-element annular array which allowed for adjustable sonication depths. The phasing was set to a fixed 30 mm sonication depth for all experiments, based on literature estimates of the scalp-cortex distance to the hand motor area (*Fox et al., 2004*; *Stokes et al., 2005*).

## Transducer characterization

To characterize the intensity of the active sonication paradigms (*Figure 3B*), the transducer was submerged in a degassed distilled water tank with the active face 30 mm from a calibrated fiber optic hydrophone (Precision Acoustics, PAFOH03, Dorchester, UK). The transducer was continuously excited at the fundamental frequency (500 kHz) in power steps of 1W, up to a maximum of 20W. The oscilloscope measurements of maximum peak voltages per cycle was multiplied by the hydrophone sensitivity (94.6923 mV/MPa) to derive a corresponding instantaneous pressure ($P_i$). The instantaneous intensity ($I_i$) was then calculated by squaring the $P_i$ and multiplying by the inverse of the density (997 kg/m$^3$) and speed of sound (1498 m/s) in the propagating medium. The pulse intensity integral (PII) is then derived by integrating the instantaneous intensity over the duration of the entire pulse. From the PII, two measures of acoustic exposure can be derived: the spatial-peak temporal average ($I_{SPTA}$), and the spatial-peak pulse average ($I_{SPPA}$). To characterize the axial intensity profile at the acoustic focus, a three-axis robotic stage moved the hydrophone at 0.2 mm increments parallel to the transducer face, recording the intensity profile within a 64 mm$^2$ plane centered at the acoustic focus. Calculated intensity values at different parameters used are found in *Table 1*:

## Electromagnetic and acoustic field brain mapping

The scalp position of each subject's TMS coil during sonication, including the 10 mm scalp-coil offset, was captured with neuronavigation based on the subject's individual structural T1-weighted MRI. After applying the Montreal Neurological Institute (MNI) atlas registration to individualized scans, the MNI stereotaxic coordinates corresponding to each stimulator position were recorded. SIMNIBS, a TMS simulation software (*Thielscher et al., 2015*) was used to create a probabilistic averaged map of the induced electromagnetic current across the cortex all participants, mapped to an MNI head model (*Figure 2A*).

To model acoustic propagation of applied TUS, each participant's coronal MRI slice corresponding to the neuronavigation-captured transducer position was manually segmented into scalp, skull, and brain tissue planes (*Hynynen and Sun, 1999*; *Rosnitskiy et al., 2019*; Appendix 1). The segmented tissue layers were treated as homogenous tissue masks with material properties such as attenuation coefficient, sound velocity, and density derived from the literature (*Mueller et al., 2017*; *Robertson et al., 2018*; *Robertson et al., 2017*). An acoustic simulation toolbox, k-Wave (*Treeby and Cox, 2010*), was used to generate simulations of the acoustic focus for each participant based on individualized transducer positions acquired via neuronavigation (*Figure 2B–C*) and the resulting pressure field was mapped back onto the original MRI images (*Figure 2—figure supplement 1*).

## TMS-TUS stimulation delivery

A custom 3D-printed holder was developed to hold the transducer to the underside of the TMS coil (*Figure 1*); the source file for 3D printing has been made freely available online (*Fomenko, 2019a*; copy archived at swh:1:rev:9e494e2f296518d4a596e5e221e3b5481cc07cda). The ultrasound transducer was rigidly fixed to the underside of the *Figure 8* coil, and held in the center of the coil, between the two windings. Previous validation studies (*Opitz et al., 2014*) showed that the measured electromagnetic maxima of a figure-of-8 TMS coil is located between the two coil windings, justifying our central placement of the transducer holder.

Single-pulse TMS delivery was via a 70-mm figure-eight coil powered by a magnetic stimulator (Magstim Company, Dyfed, UK). The handle of the coil pointed backwards and laterally at 45° from midline. At the beginning of each experimental session, we determined the FDI motor hotspot on the scalp, defined as the location over which TMS evoked MEPs of highest peak-to-peak amplitude in the target muscle at a given suprathreshold stimulator intensity (*Bashir et al., 2013*). A marker was used to precisely trace outline of the transducer holder on the scalp, to ensure that the angle and position of the wand was captured. TMS output intensity was adjusted to produce MEPs with a mean peak-to-peak amplitude of approximately 0.5 mV over 10 trials in the relaxed contralateral FDI muscle (*Hamada et al., 2008*). The direction of the induced current was from posterior to anterior and activated the corticospinal neurons transynaptically. We captured the position of the coil in stereotactic space by registering the subject's individual T1 anatomical MRI with the position of the TMS coil, using a TMS tracker and infrared camera via Brainsight.

Aqueous compressible gel pads (Aquaflex, Parker Laboratories, NJ, USA) were cut into 1.5 mm thick pads 40 mm in diameter and placed between the surface of the transducer and the subject's scalp. Any visible air bubbles at the transducer-pad interface were manually extruded, and a small amount of ultrasound gel (Aquasonic 100, Parker Laboratories, NJ, USA) was applied at the scalp-pad interface (Appendix 2). The pads were replaced between each sonication block, or approximately every 6 min.

## Sham condition

For our initial experiments aiming to reproduce the suppressive effects found with a single set of basic parameters (*Legon et al., 2018a*), we used two types of sham: active and inactive (*Figure 9*). The active sham consisted of manually flipping the transducer so that the active face was pointing away from the scalp, as described previously (*Legon et al., 2018a*). The novel inactive sham consisted of maintaining the active face of the transducer facing toward the scalp, to later allow for efficient randomization blocks of active and sham conditions without the need to manipulate the transducer between trials.

To mask the slightly audible buzz of the active piezoelectric element during sonication, a signal generator (Agilent 33220A, Keysight Technologies) delivered a tonal waveform 0.5 s in duration to a set of speakers positioned two meters lateral to the subject's left ear (*Braun et al., 2020*). The sound was tuned to match the audible frequency (10–15 kHz) of the active transducer until the subject could not distinguish the speaker noise from a 0.5 s activation of the transducer. This sound was triggered identically every time a TUS or sham condition was delivered to the transducer.

## Electromyography recording

Surface electromyography (EMG) was recorded from the right FDI muscle contralateral to the stimulated motor cortex using disposable disc electrodes with a belly-tendon montage. EMG was 1K amplified (Intronix Technologies Corporation Model 2024F, Bolton, Ontario, Canada), filtered (band pass 2 Hz–2.5 kHz), digitally sampled at 5 kHz (Micro 1401, Cambridge Electronics Design, Cambridge, UK) and stored in a personal computer for off-line analysis. Subjects were seated upright in a chair and asked to keep eyes open throughout the experimental session.

## Effects of basic TUS parameters on resting motor cortex excitability

Two blocks of 10 TMS-TUS stimuli to M1 were administered to 12 subjects, with the active transducer face coupled to the scalp for the verum TUS block, and pointing away from the scalp (sham) for the sham block. For four additional subjects, two blocks of 20 TMS-TUS stimuli were delivered, with one block consisting of verum TUS and the other inactive sham. Overall, each participant

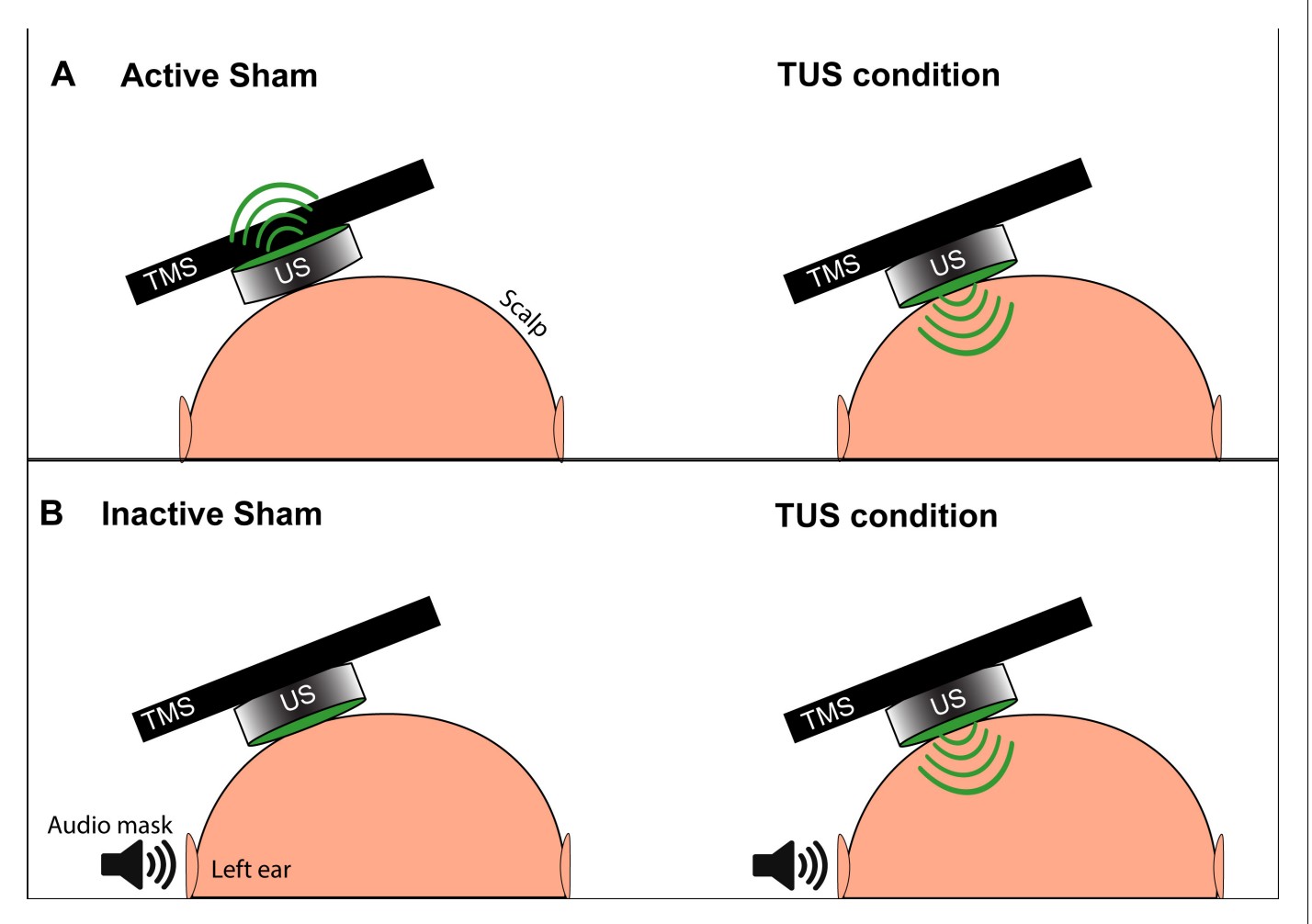

**Figure 9.** Depiction of the two types of TUS sham and masking used in experiments. (**A**) The active sham condition, as per *Legon et al., 2018b* involves flipping the active face (green) of the transducer to point away from the scalp and delivering acoustic energy away from the subject (left). During the active TUS condition, the transducer is flipped over to deliver acoustic energy transcranially (right). (**B**) Our inactive sham condition used for quick successive delivery of experimental conditions involves keeping the active face of the transducer always oriented toward the scalp (left), and only activating the transducer and delivering acoustic energy transcranially during an active TUS condition (right). This allows for interleaving experimental conditions with sham without the need to manually reposition the transducer. An audible tone lasting 0.5 s is played near the ipsilateral ear during both conditions to mask TUS delivery.

received a cumulative sonication time between 5 and 10 s. Block order was random for each subject. The interstimulus interval was 5 s, and the resultant MEPs were collected for analysis. Basic sonication parameters were similar to those in prior human reports (*Ai et al., 2016*; *Legon et al., 2018b*): fundamental frequency = 500 kHz, pulse repetition frequency (PRF) = 1000 Hz, duty cycle (DC) = 30%, sonication duration = 0.5 s, and intensity = 2.32 W/cm$^2$ I$_{SPPA}$. The TMS pulse was time-locked to 10 ms before the end of sonication. (*Figure 10*).

### Effects of different TUS parameters on resting motor cortex excitability

Four separate experiments each featuring systematic examination of a distinct sonication parameter were conducted in the following order: DC (sham, 10, 30, 50%), Sonication duration (sham, 0.1, 0.2, 0.3, 0.4, 0.5 s), PRF (sham, 200, 500, 1000 Hz), and adjusted PRF (sham, 200, 500, 1000 Hz). The fundamental frequency, intensity, and other parameters not being varied in each experiment were kept the same as the basic parameters. In the adjusted PRF experiment, the burst duration was held constant, requiring an adjusted DC (10, 25, 50%) for the PRF conditions 200, 500, 1000 Hz, respectively. For all experiments, the TMS pulse was time-locked to 10 ms before the end of sonication. For each

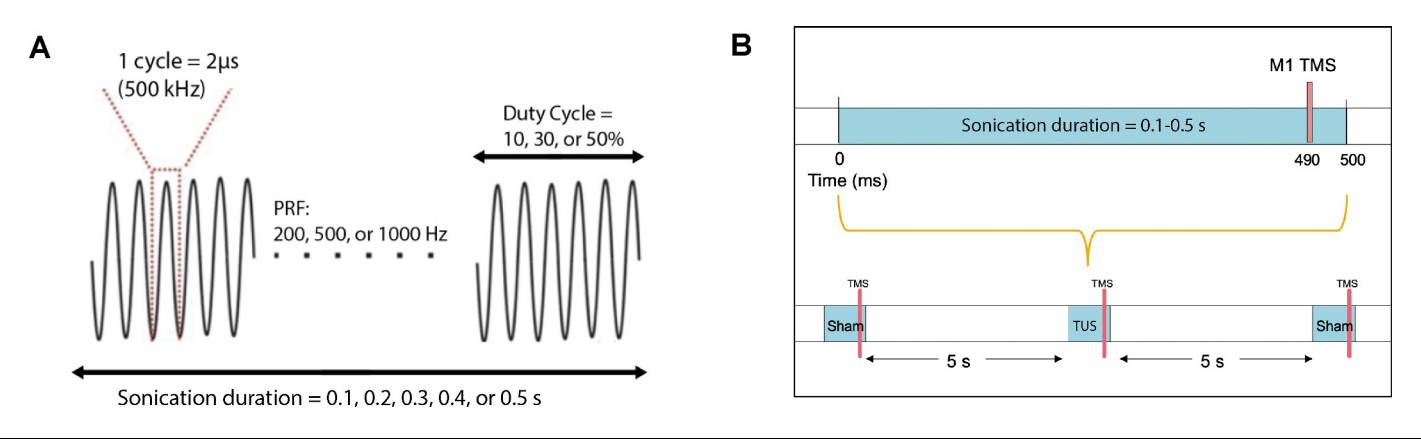

**Figure 10.** Acoustic parameters and timing relative to TMS. (**A**) Acoustic parameters of low-intensity ultrasound which were varied in the experiments include sonication duration, pulse repetition frequency, and duty cycle. The fundamental frequency was held constant at 500 kHz. (**B**) TMS and TUS stimulation delivery while varying the sonication duration parameter. Different durations (0.1, 0.2, 0.3, 0.4, 0.5 s) were randomized with the sham condition, where TUS was not delivered. TMS was always time-locked to 10 ms before the end of sonication. An inter-stimulus interval of 5 s was used between each stimulation epoch.

parameter of interest, the order of condition delivery was randomized using a custom MATLAB 2019b script controlling both the TMS and TUS delivery (*Fomenko, 2019b*; copy archived at swh:1: rev:fe4df0a5372942406324efa09ec93fe2192487ea). For 12 participants, 10 TMS-TUS stimuli per condition were delivered in randomized order with an interstimulus interval of 5 s, and the resultant MEPs were collected for analysis. For four additional subjects, 15 stimuli were collected per condition. Each participant received a cumulative sonication time between 60 and 90 s. For all conditions, the active face of the transducer was coupled to the scalp, and the masking sound was played for 0.5 s on the speaker. For the sham condition, the transducer was not excited, but the masking sound was played identically to verum parameters.

## Effects of TUS on active MEP and silent period

The maximum voluntary contraction (MVC) was acquired by noting the oscilloscope display output, which was connected to the EMG amplifier through a leaky integrator while the subject squeezed a plastic cube between the right thumb and index finger with maximal effort. For the active conditions, we recorded EMG during 20% of MVC of the right FDI while stimulating the left motor cortex at 120% of resting motor threshold (RMT). The RMT was defined as the minimum stimulation intensity that could elicit MEP of no less than 50 μV in 5 out of 10 trials when the participant was fully relaxed. The silent period duration was measured from the onset of MEP to the reoccurrence of any background EMG activity according to previously described methods (*Farzan et al., 2013*). TMS was delivered 10 ms before the end of sham or active TUS, and the sonication parameters were the same as the *basic parameters* described above. Each participant received 15 trials each of sham and TUS, with a cumulative sonication time of 7.5 s.

## Effects of TUS on intracortical inhibition and facilitation tested with paired-pulse TMS

The outputs of two Magstim 200 stimulators were directed to a Bistim$^2$ module (Magstim Company, Dyfed, UK), which was connected to the 70 mm TMS coil. The timing of the pulses was controlled by the output features of the A/D converter (Micro 1401, Cambridge Electronics Design, Cambridge, UK). The sonication parameters were the same as the *basic parameters* described above. The first TMS stimulus was delivered 10 ms before the end of TUS. To ensure that any differences in paired-pulse inhibition or facilitation between TUS and sham are not due to a TUS-mediated reduction of test stimulus (TS) MEP amplitude, but rather to differences in susceptibilities of pathways tested by the paired TMS paradigms to TUS, we further measured the RMT when delivering with sham and

TUS, and adjusted the stimulator intensity to account for the effects of TUS. For each distinct paired-pulse paradigm (i.e.: SICI), the number of trials per participant were as follows: TS alone: 20 trials (10 TUS, 10 sham); conditioning stimulus (CS)-TS: 20 trials (10 TUS, 10 sham). Each condition block consisted of 20 trials delivered in random order, and each participant received eight paired-pulse blocks, with a cumulative sonication time of 40 s.

For the SICI and ICF experiments, we delivered a subthreshold CS at 80% RMT and the TS intensity was at the intensity that elicited MEP of ~1 mV. The interval between CS and TS was 2 ms for SICI and 10 ms for ICF. For the LICI, both the CS and TS consisted of an identical suprathreshold intensity able to evoke mean MEPs of ~1 mV. For SICF, the first stimulus intensity was set to generate MEP of ~1 mV. The second stimulus was at RMT. The interval between stimuli was 1.5 ms for the first peak and 2.9 ms for the second peak of SICF.

## Effects of TUS on a visuo-motor task

To measure the effects of TUS on voluntary motor behavior, we used a visuo-motor task similar to one previously used to assess motor performance and reaction time after electrical deep brain stimulation (*de Almeida Marcelino et al., 2019*). Each participant was seated in a comfortable chair facing a computer screen, with an eye-to-screen distance of 60 cm. A three-axis accelerometer was attached to the dorsal aspect of the distal right index finger and calibrated so that abduction of the FDI triggered a voltage rise in the horizontal plane of the accelerometer. The ultrasound transducer was positioned at the previously-marked FDI hotspot over the left M1, coupled with a gel pad, and secured to the head using a 3D-printed holder with Velcro straps (*Fomenko, 2019a*).

Participants were instructed to steer a cursor using their index finger on a digital tablet (Intuos Draw, Wacom) as quickly and accurately as possible to target circles appearing in one of three on-screen positions (*Figure 8—figure supplement 1*). Participants were asked to only abduct or adduct the index finger, while keeping the hand, arm, and remaining digits still. Trials started with a red circle at the starting location (diameter = 50 pixels) that turned green after 2 s to indicate to the participants that they should move to the target location. The target location was indicated with a blue circle (diameter = 200 pixels) with a white cross at the center, which appeared simultaneously as the start location turned green. The target location appeared at one of three randomized locations with 60 trials for each: (1) close target, 600 pixels away, (2) medium target, 1200 pixels, or (3) far target, 1800 pixels away. On each trial, TUS or sham was applied 250 ms before the blue circular target appeared on the screen for a total sonication duration of 500 ms. TUS was applied either at full intensity (2.32 W/cm$^2$), as with the single- and paired-pulse experiments, or at a close to somatosensory threshold acoustic intensity (0.54 W/cm$^2$). The near-threshold intensity was chosen to correspond to a lack of subjective scalp tactile sensation in all participants after five test sonications were applied at basic parameters. For both TUS and sham conditions, the masking sound was played over a speaker in a similar fashion as the TMS experiments. Subjects first practiced the task until performance speed plateaued, after which the transducer was positioned on the scalp and 180 trials pseudorandomized across TUS/sham conditions (90 trials each) were performed, with a cumulative sonication time of 45 s. MATLAB 2019b was used to record all finger traces via a modified Psychophysics Toolbox script and Spike2 (Cambridge Electronics Design, Cambridge, UK) was used to record the accelerometer trace (*Brainard, 1997*; *Fomenko, 2019b*).

## Statistical analysis

For the MEP data, a custom script in SIGNAL (Cambridge Electronics Design, Cambridge, UK) was used to extract peak-to-peak MEP amplitudes, and to extract the median amplitude by condition. PRISM 8 (GraphPad Software Inc, San Diego, California) was used for statistical analysis. For the single-pulse basic parameters and active MEP experiments, two-tailed paired t-tests were used to compare means of participant MEP amplitudes between verum TUS and sham conditions. For the interleaved and blocked systematic parameter variation experiments, MEP mean amplitudes were first analyzed using repeated-measures (RM) one-way ANOVA. Post-hoc two-tailed paired t-tests were then used to compare each parameter to its respective sham group, and p-values were adjusted for multiple comparisons using Holm's sequential Bonferroni procedure (*Holm, 1979*), with omnibus significance α = 0.05. Analysis of blocked experiments by trial number was performed with multiple two-tailed paired t-tests, between the means of the first three trials (first 20%) and the last

three trials (last 20%) of all participants, for each block of 15 total trials. Paired-pulse experiments were expressed as the ratio of the conditioned (with preceding CS) to the unconditioned (TS alone) MEP amplitudes, for both sham and TUS, and two-tailed paired t-tests were used. For the behavioral task, reaction time was defined as the time from target stimulus presentation to the horizontally-directed accelerometer voltage trace exceeding two standard deviations from baseline. Total path travelled was defined as the number of pixels contained within the path from start to finish, while deviation score was the deviation in pixels from an ideal straight line from start to finish target. Mean values of TUS and sham were analyzed with two-tailed paired t-tests.

## Acknowledgements

We thank Cricia Rinchon and Julianne Baarbé for their help in a piloting the optimal coil-scalp distance for this experiment. Dr. Neil M. Drummond is thanked for constructive discussion and assistance. We thank Dr. Suneil K Kalia and Dr. William Hutchison for their guidance during committee meetings. Dr. Ryan M Jones is thanked for his advice regarding acoustic simulations, and we are grateful for Frank Vidic's electrical expertise relating to devices used in the project. This work was supported by the Canadian Institutes for Health Research (CIHR) Foundation Grant (FDN 154292, RC), Banting and Best Doctoral Award (AF), the Clinician Investigator Program – University of Manitoba (AF), and the Canada Research Chair in Neuroscience (AML)

## Additional information

### Funding

| Funder | Grant reference number | Author |
| --- | --- | --- |
| Canadian Institutes of Health Research | Banting and Best Doctoral Award | Anton Fomenko |
| NSERC | Discovery Grant RGPIN-2020-04176 | Robert Chen |
| Canadian Institutes of Health Research | Foundation Grant FDN 154292 | Robert Chen |
| University of Manitoba | Clinician Investigator Program | Anton Fomenko |
| Canada Research Chairs | Neuroscience | Andres M Lozano |
| University Health Network | R.R. Tasker Chair in Stereotactic and Functional Neurosurgery | Andres M Lozano |

The funders had no role in study design, data collection and interpretation, or the decision to submit the work for publication.

### Author contributions

Anton Fomenko, Conceptualization, Data curation, Formal analysis, Funding acquisition, Investigation, Visualization, Methodology, Writing - original draft, Project administration, Writing - review and editing; Kai-Hsiang Stanley Chen, Conceptualization, Resources, Data curation, Formal analysis, Funding acquisition, Investigation, Methodology, Project administration, Writing - review and editing; Jean-François Nankoo, Conceptualization, Investigation, Methodology, Writing - review and editing; James Saravanamuttu, Data curation, Formal analysis, Investigation, Writing - review and editing; Yanqiu Wang, Investigation, Writing - review and editing; Mazen El-Baba, Software, Visualization; Xue Xia, Investigation; Shakthi Sanjana Seerala, Resources, Investigation; Kullervo Hynynen, Resources, Writing - review and editing; Andres M Lozano, Robert Chen, Conceptualization, Supervision, Writing - review and editing

### Author ORCIDs

Anton Fomenko https://orcid.org/0000-0003-4131-6784

## Ethics

Human subjects: All patients gave written informed consent and the protocol was approved by the UHN Research Ethics Board (Protocol #18-5082) in accordance with the Declaration of Helsinki on the use of human subjects in experiments.

## Decision letter and Author response

Decision letter https://doi.org/10.7554/eLife.54497.sa1
Author response https://doi.org/10.7554/eLife.54497.sa2

## Additional files

### Supplementary files

• Transparent reporting form

### Data availability

Data used for this study are included in the manuscript and supporting files. Files for 3D printing the stimulating devices and custom MATLAB scripts used for stimulation have been deposited into a cited GitHub repository.

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

## Appendix 1

### Acoustic simulation methods

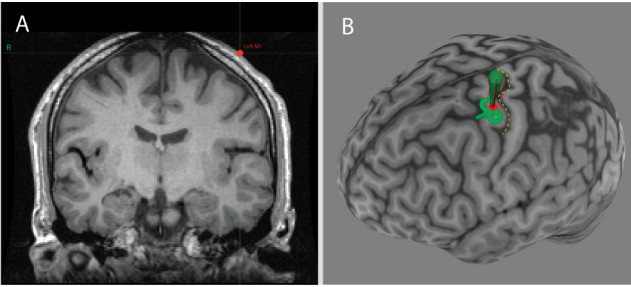

**Appendix 1—figure 1.** Neuronavigation software (Brainsight) showing the individualized location of the US transducer on the scalp captured during US/TMS stimulation of one participant. Panel A: Coronal T1-weighted MRI of the brain at the location of the US transducer (red dot). Panel B: Three-dimensional reconstruction of the participant's cortical surface, with the central sulcus highlighted with a yellow dashed line. The red dot represents the scalp location of the transducer and the green path represents the predicted path of sonication perpendicular to the transducer face.

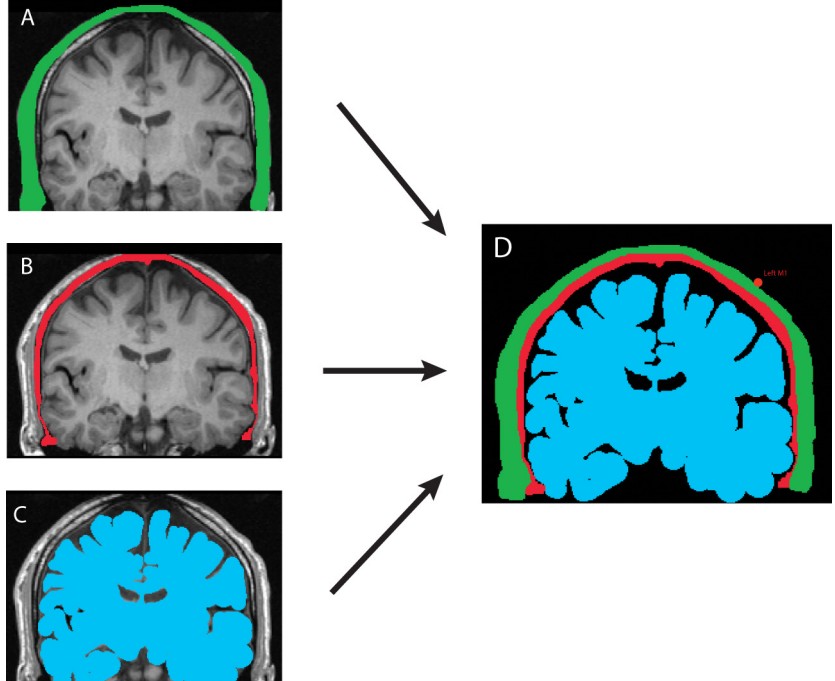

**Appendix 1—figure 2.** MRI-based manual tissue segmentation pipeline shown for a single participant. Tissue properties were set as follows, with a medium alpha power (y) = 1.43. Panel (**A**) Skin and scalp (green). Alpha coefficient: 2.05 dB/(MHz $^y$ cm), sound speed: 1732 m/s, density: 1100 kg/m$^3$. Panel (**B**) Bone (red). Alpha coefficient: 8.83 dB/(MHz $^y$ cm), sound speed: 2850 m/s, bulk density: 1732 kg/m$^3$. Panel (**C**) Brain (blue). Alpha coefficient: 1.00 dB/(MHz $^y$ cm), sound speed: 1552 m/s, density: 1040 kg/m$^3$. Panel (**D**) The three masks were used to set the properties of the acoustic propagation medium in the subsequent numerical simulation. The background medium was set as water (black) with the following properties: Alpha coefficient: 0.05 dB/(MHz y cm), sound speed: 1482 m/s, density: 1000 kg/m$^3$.

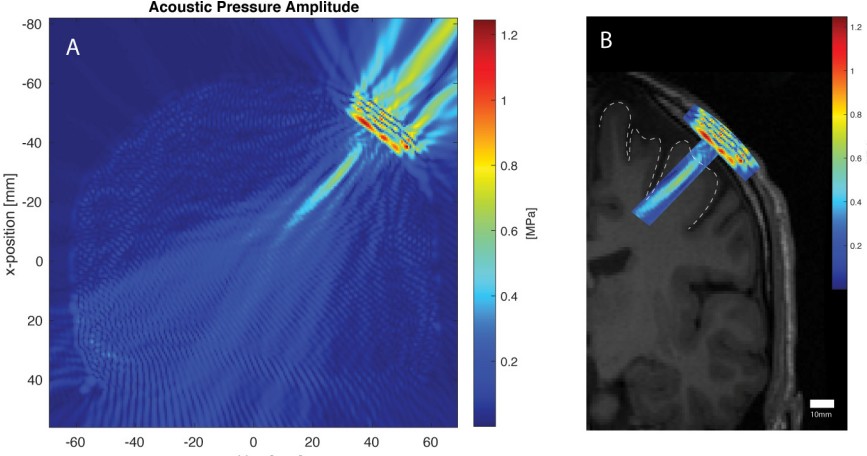

**Appendix 1—figure 3.** Acoustic simulation and visualization for the same participant using the k-WAVE Matlab toolbox. The parameters of the simulation were as follows: Fundamental frequency: 500 kHz, Grid points: 472 × 472, Points-per-wavelength: 10, Points per period: 67, Courant-Friedrichs-Lewy number: 0.15, Size of perfectly matched layer: 20 grid points. Panel A: Acoustic pressure profile of simulated 500 kHz transducer sonicating against the participant's scalp at the scalp location determined by neuronavigation. Cigar-shaped focus within the brain, and artifact from non-active transducer face can be seen. Panel B: Pressure profile deep to the transducer overlaid onto participant's T1 MRI (left hemisphere only shown), with the primary motor cortex cortical surface shown with a dotted line. Areas outside the acoustic focus are cropped.

## Appendix 2

## Validation of air-free interfaces

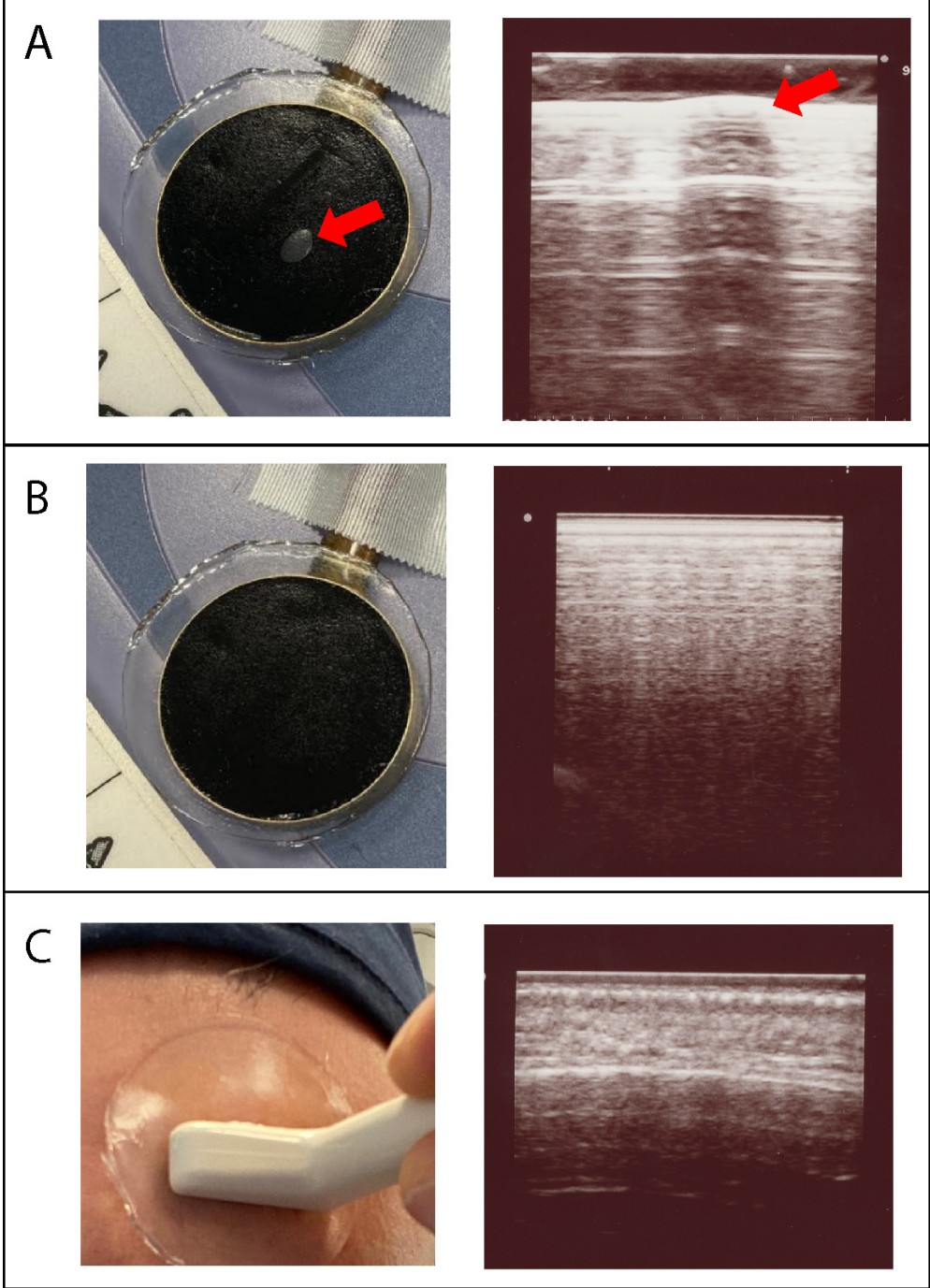

**Appendix 2—figure 1.** Visual and imaging validation of air-free coupling interfaces. Panel (**A**): Visible air bubble at the TUS transducer-gel interface, indicated by the red arrow (left), and corresponding refraction and shadowing artifact seen when an imaging transducer (Hitachi Aloka ProSound Alpha 7) is placed over the air bubble (right). Panel (**B**) Manual smoothing of the gel pad is performed until any small bubbles are extruded, leaving a homogenous black interface (left), and confirmatory imaging with ultrasound probe at the gel pad surface shows no artifact at the TUS

*Appendix 2—figure 1 continued on next page*

*Appendix 2—figure 1 continued*

transducer-gel interface (right). Panel (**C**) Application of gel pad over the frontal bone with the imaging transducer applied over the gel, visualizing the scalp-pad interface (left). Imaging of the scalp-pad interface, showing underlying tissue layers with no visible artifacts (right).

