## [Decision Letter]

**Acceptance summary:**

In their study, Fomenko and colleagues measure the effects of transcranial ultrasound stimulation (TUS) on corticospinal excitability changes as assessed by transcranial magnetic stimulation (TMS) and motor evoked potentials (MEP) in primary motor cortex. This study is very timely as only a few TUS studies have been published in humans so far. Importantly, they systematically investigated the impact of various TUS parameters and TMS pulses on cortical excitability.

**Decision letter after peer review:**

Thank you for submitting your article "Systematic examination of low-intensity ultrasound parameters on human motor cortex excitability and behaviour" for consideration by *eLife*. Your article has been reviewed by Richard Ivry as the Senior Editor, Laura Dugué as the Reviewing Editor, and three reviewers. The following individual involved in review of your submission has agreed to reveal their identity: Wynn Legon (Reviewer #3).

The reviewers have discussed the reviews with one another and the Reviewing Editor has drafted this decision to help you prepare a revised submission.

Summary:

Fomenko et al., combined transcranial ultrasound stimulation (TUS) and transcranial magnetic stimulation (TMS) of the primary motor cortex (M1) in humans to measure the effects of TUS on corticospinal excitability changes as assessed by TMS and measures of motor evoked potentials (MEP). They systematically investigated the impact of important TUS parameters, such as pulse repetition frequency, duty cycle, and sonication duration on several single- and paired-pulse TMS related indices for GABA-A- (SICI) and -B-receptor mediated inhibition (LICI, cortical silent period) as well as facilitation (SICF, ICF). They found corticospinal excitability to be generally decreased by TUS, and stronger so for longer sonication and shorter duty cycles, as well as a stronger SICI during TUS. They also observed shorter response times during TUS in a visuomotor task. No impact of TUS was found on any of the other parameters.

The reviewers and the editors appreciates the need for such methods paper as TUS is a very promising neurostimulation technique that is at the stage of translating from animal to human research. Despite a few (<10) studies being already published in humans, the impact of the most basic TUS parameters on cortical excitability is still unclear, and systematic studies like this one are very much needed.

That said, there are significant methodological and statistical concerns that need to be addressed before the paper can be published. The following comments highlight the "essential revisions.

Essential revisions:

1) In general, the data lacks apparent robustness. They authors should collect data from more participants (preferably with more trials) to allow for correction for multiple comparisons across all the tested indices and solve the apparent lack of robustness (failed within-study replications) of the data. The following points describe the different results in which robustness issues have been noticed.

1a) Given the relatively low number of subjects and a total of ~16 different measurements being investigated, there is a certain risk of false positive results. Do the results survive correction for multiple comparisons?

1b) Several of the experiments that either vary TUS parameters or investigate paired-pulse effects also contain a single-pulse MEPS with TUS at "basic parameters" for which a clear suppressive effect was found (p = 0.0018) in the beginning. However, in those four experiments, this effect does not seem to replicate: Figure 5C: DC of 30%; Figure 5E: Pulse repetition frequency of 1000 Hz; Figure 7: TS; Figure 7: S1.

1c) Statistical analyses for more than two conditions seem based on one-way rm-ANOVAs (such as 5 different sonication durations normalized to sham), "and conditional on a significant f-value, Dunnett's multiple comparisons was performed to explore groups with significant differences from sham" (subsection “Statistical analysis”). The Dunnett's tests presumably test something very different (namely the differences between each TUS condition against Sham) than the ANOVA, which testes for differences of the sham-normalized conditions with respect to each other, and not of the conditions relative to sham. (a) The Dunnett's test should thus not be conditional on the f-test, as they simply answer very different questions. This would change if the ANOVA would based on raw MEP values and include the sham condition as level. (b) This also means that there is no post-hoc evaluation of the differences between conditions, which would require e.g., post-hoc paried t-tests. (c) Potentially, some legitimate basic comparisons against sham have not been performed, because the f-test was non-significant, even though it tested something very different.

2) The MEP analysis and results need to be reported with more details. The following points should be clarified in the revised version of the manuscript.

2a). It is unclear how many MEPs (for paired-pulse, how many CS+TS and TS alone trials, respectively) were acquired per condition of each of the experiments. It reads like only 10 MEPs were acquired per condition (20 for paired-pulse blocks, so again 10 CS+TS and 10 TS alone). Given the known high variability of the MEP (cf. also subsection “Limitations” "the MEP variability we saw in some subjects") and the low number of participants (N = 12; which is understandable for TUS in humans but still low) these low MEP numbers are problematic. A larger number (20 or more MEPs) per condition would provide a much more stable estimate and allow the detection of small effects. Although there are quite some clinical neurophysiology papers out there with 10 or 15 MEPs, the kind of results presented here may shape the translation of TUS for human application and are thus too important for the community to suffer from low statistical power. Some of the findings (also of the negative ones) are in contrast to a previous study (Legon et al., 2018), and it is unclear whether this has to be attributed to differences in experimental design or simply noise. Many separate measures were obtained for the same subjects, and the effort of the authors is acknowledged, but maybe less measures and more trials would have been the better choice? Please report to point 1.

2b) The authors used the procedure of outlier exclusion. Outlier removal is a controversial method for MEP analyses. MEP amplitudes are not normally distributed but rather follow a power law, and removal of extremes is therefore corrupting the data. Was the pre-activation controlled and excluded together with the outliers? How many MEPs were left for the statistical analysis after exclusion? How do the results change when all trials are kept but the median is used per condition (instead of the mean) or the mean of log-transformed MEP values? Given that robustness is key for these kind of results, transformations should be avoided (or outlier removal at least).

3) Several concerns have been raised regarding M1 targeting and the use of a normalized brain and MNI coordinates. Specifically, in the newly collected data, M1 (or specific parts of it) should be properly targeted by using neuronavigation and individual head models with T1/T2 (instead of a normalized brain and MNI coordinates). Moreover, the following points should be addressed in the revised version of the manuscript.

3a) Why was a normalized brain used? You have a neuronavigation system and each participants' MRI. Why MNI coordinates? Should you not distinguish based on each participants' anatomy and TMS response to determine location?

3b) Why was a mark placed on the head when you have a neuronavigation system? Since BrainSight neuronavigation was used to identify the TUS transducer position on the scalp, why was it not used to ensure and maintain correct transducer placement throughout the many measurements and sessions? Given the small diameter of the sonication beam, tiny changes in tilt or position can have a massive effect on the actually stimulated part of cortex.

3c) Figure 2B: Please also provide sagittal and axial views to allow a better judgement of the targeting of M1. Is it actually targeting M1 or maybe premotor cortex? Which part of the precentral gyrus is actually sonicated?

3d) According to Fox, 2006 and Geyer, 1996 the motor cortex of human is allocated in the sulcus and at best to a small extent at the crown. In Figure 2B the white matter is targeted as well as in Figure 3A where the pyramidal neurons are allocated in the white matter. Thus, the first part of the following sentence might not be accurate: "Similarly, the individual simulations of ultrasound propagation for each participant confirmed acoustic targeting of a portion of M1, as well as underlying white matter tracts. (Figure 2B)." see also "Our finding may suggest that cortical interneurons in layers II/III which are well-encompassed within the acoustic focus (Figure 4)"

3e) Without a CT scan and only T1-weighetd images no really reliable simulations can be obtained for the acoustic waves. Figure 2B only shows one "representative" subject. Have simulations been performed for all subjects or was the transducer only placed on top of the TMS M1 hotspot for each subject without modelling the sonication beam individually? This assumes that the relevant motor neurons of M1 are actually directly beneath the coil center, which is not necessarily the case.

3f) Figure 3A: The location and orientation of corticospinal output neurons in M1 is incorrect and misleading. They are actually located in the anterior bank of the central sulcus and oriented tangentially to the scalp. This should be corrected.

4) The phrasing "… a portion of M1…" is disconcerting. Because the transducer was concentric with the intersection of the TMS coil, wherever you put the coil is where the US was. Is that accurate? How did you confirm this however in your models if you did not use individual MRI but rather normalized MRI. MNI coordinates are mentioned previously but not given anywhere in the manuscript. TUS is highly localized and using generalized MNI coordinates is not appropriate. Please provide an acoustic wave modeling of the sonication for each participant.

5) There is no data on how/if the transducer affected the TMS pulse or vice versa. This needs to be either collected or cited from Legon et al., 2018 and differences in transducer materials/design should be factored in if there are any.

6) Gel pads are notorious for trapping air bubbles between interfaces. This can be easily detected using imaging mode of your transducer. Was this checked for? Was any other coupling media used?

7) TMS was applied in order to measure the cortical excitability changes with MEP. The TMS pulses were locked to the end of FUS or sham stimulation and they had an interstimulus interval of 5 seconds. If there was no jitter for TMS pulses it means that rTMS at 0.2 Hz was applied simultaneously with FUS. Repetitive TMS applied at a very low frequency of 0.2 Hz has been shown to be effective in several studies (Urushihara, 2006; Hosono, 2008). For example, rTMS over PMC led to an increase in somatosensory evoked potentials. Could the possible effect of low frequency rTMS on cortical excitability when applied simultaneously with FUS be discussed?

8a) A further concern is "This sound was triggered every time a FUS or sham condition was delivered to the transducer." This could mean that the effects reflect acoustic TMS pairing, see e.g. doi: 10.3389/fnhum.2014.00398, other papers are around as well. Can the results in Figure 5D be due to longer acoustic stimulation? I expect the sham condition to be performed with the shortest duration, however not sure? Can this explain the lack of an effect in Figure 5E and F?

8b) Also: "and the task was more complex; nevertheless, the sonication parameters and cortical location were similar, and we observed an effect size of about 100 ms, though with higher variability." It may simply be that the start of the sonification sound leads via a kind of pre-triggering to shortened responses. This is discussed by the authors in subsection “Behaviour”. The effect in Figure 8A appears to be implausibly high. Control experiments seem to be reasonable with very light somatosensoric or close to threshold acoustic stimulation. The whole field of TMS-EEG suffers from acoustic and somatosensory contamination.

[Editors' note: further revisions were suggested prior to acceptance, as described below.]

Thank you for resubmitting your article "Systematic examination of low-intensity ultrasound parameters on human motor cortex excitability and behaviour" for consideration by *eLife*. Your revised article has been reviewed by Richard Ivry as the Senior Editor, a Reviewing Editor, and two reviewers. The following individuals involved in review of your submission have agreed to reveal their identity: Til Ole Bergmann (Reviewer #2); Wynn Legon (Reviewer #3).

The reviewers have discussed the reviews with one another, and the Reviewing Editor has drafted this decision to help you prepare a revised submission.

Summary:

Fomenko et al., have thoroughly revised their manuscript. They added four new participants (now N = 16) with more trials (15-20 instead of 10 per condition), as well as more TUS modelling information. While this is not a large increase in sample size, the authors' efforts are appreciated. However, both the reviewers and the editors were concerned by the fact that the effects are robust for the 'block' experiment but not for the 'parameter' (or interleaved) experiment. Consequently, additional data should be collected in a sufficiently large sample (eg. keeping the N = 16 for interleaved and adding N = 16 for a blocked variation of stimulation parameters), targeting specifically the block vs. interleaved question for the duty cycle and pulse repetition frequency parameters.

Essential revisions:

1) Regarding the block vs. interleaved question:

If it makes indeed a difference for single trial MEP modulation whether TUS is applied in a block design with fixed parameters or with trial-by-trial variation of parameters, this would have important implications for TUS application and thus needs additional data collection, as well as detailed discussion. Specifically, for some parameters (sonication duration) the default parameter (0.5s) was effective in the interleaved design. However, for others (duty cycle) it was slightly different (10% instead of 30%), and for yet others (pulse repetition frequency) it was not different from sham anymore at all (but was previously with 1000 Hz). The blocked vs. interleaved interpretation is post-hoc and potentially valid only for duty cycle and pulse repetition frequency.

2) The authors added 4 subjects to the existing data set (N = 12) but the reviewers fail to see the significance of this. First, data from the original 12 is presented in Figure 4 and the data from the new N = 4 is presented on its own. Could the authors explain the added value of this? It looks from the manuscript that N = 16 was inclusive for the parameter testing but not the 'basic parameter' testing. The authors could either remove Figure 4C or include that data in Figure 4A.

3) It is still unclear how many MEPs were left for each participant for the statistical analysis after exclusion. It looks like the new data uses 15 trials but the old 10 and the block experiment 20. Furthermore, it is stated in the Materials and methods that MEPs {plus minus} 2SD of mean were excluded. Please include M+/-SD in the Materials and methods.

4) The following questions still need to be addressed:

- Was the pre-activation controlled and excluded together with the outliers?

- How do the results change when all trials are kept but the median is used per condition (instead of the mean) or the mean of log-transformed MEP values? Also, do the results hold when not removing the outliers?

- Why was the neuronavigation system not used to online maintain coil/transducer position? Coil angle cannot be reliably inferred from a felt pen drawing on the scalp.

5) Could the authors show the sonication beam model for all three slices (coronal, sagittal, axial)? The axial slices do not seem to be the most helpful ones when determining whether M1 was hit and which portion of it.

[Editors' note: further revisions were suggested prior to acceptance, as described below.]

Thank you for resubmitting your article "Systematic examination of low-intensity ultrasound parameters on human motor cortex excitability and behaviour" for consideration by *eLife*. Your revised article has been reviewed by Richard Ivry as the Senior Editor, a Reviewing Editor, and one reviewers. The following individual involved in review of your submission has agreed to reveal their identity: Til Ole Bergmann.

The Reviewing Editor has drafted this decision to help you prepare a revised submission.

We thank the authors for addressing the comments raised in the review process.

The new results demonstrate that the effectiveness of TUS parameters does indeed depend on whether TUS parameters are varied across trials or kept constant within a block. These additional findings are important and will be very useful for others using this approach. These results though also point to one important question that we believe should be in the published study: Are the results in the block design condition due to cumulative effects? That is, is there a systematic change over time, indicative of a cumulative effect? (To quote the reviewer who noted this concern, "Specifically, is there a build-up of suppression across trials within a block? This could be tested e.g. by regressing the MEP amplitude based on within-block trial number or by comparing early and late MEPs within a block. It would be a very important outcome to know whether the observed suppression is instantaneous or accumulating across trials.").

---

## [Author Response]

Essential revisions:1) In general, the data lacks apparent robustness. They authors should collect data from more participants (preferably with more trials) to allow for correction for multiple comparisons across all the tested indices and solve the apparent lack of robustness (failed within-study replications) of the data. The following points describe the different results in which robustness issues have been noticed.

We thank you for considering our manuscript and recognizing the need for systematic studies examining basic sonication parameters on human neural circuits. We want to thank the reviewers for their insightful suggestions and comments, which have led to a more clear and comprehensive revised manuscript. Our detailed point-by-point replies to the reviewer comments follow below.

A significant concern the reviewers raise is the limited number of subjects (n=12) in our study, and the small number of trials per condition. We thank the reviewers for giving us sufficient time to recruit 4 new healthy participants, bringing the total sample size to 16, in line with other within-subject human experiments studying TMS-elicited motor excitability and behaviour (doi: 10.1212/WNL.62.1.91, 10.1016/j.clinph.2007.09.062)

We have conducted additional modified experiments, as per the suggestion in point 2a to “reduce the number of measures and increase the number of trials”. With the 4 new subjects, we have focussed their experimental visit on testing only the five key sonication parameters, with an increase on the number of trials per parameter. Specifically, after obtaining an MRI for the new participants, we conducted the following experiments.

A) Basic parameters: Inactive Sham/TUS. 20 trials per condition, block design. 40 trials total. (Previously was 10 trials/condition).

B) Duty Cycle: Sham/10%/30%/50%. 15 trials per condition, randomized. 60 trials total. (previously was 10 trials/condition)

C) PRF: Sham/300Hz/500Hz/1000Hz. 15 trials per condition, randomized. 60 trials total (previously was 10 trials/condition)

D) Adjusted PRF: Sham/300hz/500Hz/1000Hz. 15 trials per condition, randomized. 60 trials total. (previously was 10 trials/condition)

E) Sonication duration: Sham/0.1s/0.2s/0.3s/0.4s/0.5s. 15 trials per condition, randomized. 90 trials total (previously was 10 trials/condition)

F) Near-threshold behavioural task control (See Essential revision 8b for details)

As the reviewers pointed out, auditory confounds are increasingly relevant in the TUS-neuromodulation literature, and sham conditions need to be carefully balanced and rationalized.

Accordingly, the new participants were subjected to a modified experiment A), where we have compared basic US parameters with an inactive audio-masked sham condition. The purpose of this was to validate that the type of sham we use for the rest of the study results in similar effects on the MEP as the active sham used in the original experiment A), which was also used in Legon et al., 2018).

We have created a new figure (Figure 9) explaining in detail the two types of sham, and the results are summarized in the newly-created Figure 5. For the remainder of the experiments (B-E), the inactive audio-masked sham was used, and the results were pooled with the original 12 participants, since the protocol remained the same (but with more trials).

1a) Given the relatively low number of subjects and a total of ~16 different measurements being investigated, there is a certain risk of false positive results. Do the results survive correction for multiple comparisons?

In the case of multiple conditions being compared to an independent sham condition (ie: 5 sonication durations versus sham etc.), we now use the Holm-Bonferroni method of multiple-comparison correction to perform post-hoc paired two-tailed t-tests following the ANOVA. Our original results survive correction for multiple comparisons, with the exception of the sonication duration condition “0.3 s”, which becomes non-significant when adjusted.

Materials and methods

“MEP means were first analyzed using repeated-measures (RM) one-way ANOVA. Post-hoc two-tailed paired t-tests were then used to compare each parameter to its respective sham group, and p-values were adjusted for multiple comparisons using Holm’s sequential Bonferroni procedure (Holm, 1979), with omnibus significance α = 0.05.”

1b) Several of the experiments that either vary TUS parameters or investigate paired-pulse effects also contain a single-pulse MEPS with TUS at "basic parameters" for which a clear suppressive effect was found (p = 0.0018) in the beginning. However, in those four experiments, this effect does not seem to replicate: Figure 5C: DC of 30%; Figure 5E: Pulse repetition frequency of 1000 Hz; Figure 7: TS; Figure 7: S1.

After pooling the single-pulse parameter trials from the 4 additional participants, we show results consistent with the original set of participants (See revised Results and Figure 5, Figure 6). We did not have specific data to draw a conclusion about why although the “basic” combination of US parameters suppressed cortical activity, and while some parameters replicated this result (ie: Sonication Duration), other parameters did not individually replicate when varied individually (notably PRF). We discuss potential reasons in the revised manuscript:

Discussion

“Interestingly, our results show that although a particular combination of parameters can robustly suppress TMS-elicited cortical activity when applied in a sham-controlled block design (Figure 5), parameters such as PRF when randomized and varied individually do not have the same robust effect (Figure 6C-D). From animal studies on TUS parameter dependence, PRF may be related to non-linear piezoelectric characters of the neural membrane capacity, while higher duty cycle or longer burst duration do not necessarily elicit neural activation more efficiently than the same parameter with lower magnitude (H. Kim et al., Brain Stimulation 2014). Furthermore, recent large animal studies reveal a bidirectional neuromodulation effects of varying TUS parameters (Yoon et al., 2019). Based on our results, we speculate that some PRFs may provide opposing, and potentially longer-lasting neuromodulation effects, and randomizing these stimulation blocks in succession might lead to spill-over effects from one trial to another. In contrast, the range of sonication durations tested in our experiment may have been those conducive to suppression and resulted in a robust and dose-dependent effect seen in our experiment (Figure 6B). However, this hypothesis needs further investigation.”

Regarding the TS and S1 results (Figure 8), the apparent lack of suppression seen in the TUS condition compared to sham was because of a stimulator adjustment in the calibration stage of the experiment to compensate for the robust suppressive effects of US. Before adjustment, basic parameters did in fact replicate the suppression previously seen (Figure 8, first panel), and we now add a paired t-test (p=0.016) showing that a significant mean increase of about +1.3% in stimulator intensity is required to elicit the same 1 mV average MEP values. We also now clarify in the caption that the TS/S1 MEP amplitudes were performed after this adjustment.

In paired-pulse TMS studies (e.g. SICI, SICF), It is known that the MEP amplitude induced by TS affects their inhibitory or facilitatory results (Sanger et al., 2001; Peurala et al., 2008; Ni et al., 2013). We found from our single-pulse experiments, and those of Legon, (2018) that US generally suppresses MEPs.

Therefore, we needed to ensure that any differences in paired-pulse inhibition/facilitation between FUS/sham are not due to a FUS-mediated reduction of TS/S1 MEP amplitude, but rather to differences in susceptibilities of pathways tested by the paired TMS paradigms to FUS. We clarify this rationale in the revised manuscript in the Materials and methods.

1c) Statistical analyses for more than two conditions seem based on one-way rm-ANOVAs (such as 5 different sonication durations normalized to sham), "and conditional on a significant f-value, Dunnett's multiple comparisons was performed to explore groups with significant differences from sham" (subsection “Statistical analysis”). The Dunnett's tests presumably test something very different (namely the differences between each TUS condition against Sham) than the ANOVA, which testes for differences of the sham-normalized conditions with respect to each other, and not of the conditions relative to sham. (a) The Dunnett's test should thus not be conditional on the f-test, as they simply answer very different questions. This would change if the ANOVA would based on raw MEP values and include the sham condition as level.

We thank the reviewer for this suggestion. We have now removed the stipulation that a significant F-test is required to perform post-hoc analysis in the revised manuscript. Based on this and other reviewer comments, we now show raw mean MEP amplitudes instead of sham-normalized ratios, allowing for direct comparison with the sham group.

(b) This also means that there is no post-hoc evaluation of the differences between conditions, which would require e.g., post-hoc paried t-tests. (c) Potentially, some legitimate basic comparisons against sham have not been performed, because the f-test was non-significant, even though it tested something very different.

Based on this suggestion, we now perform post-hoc paired t-tests for every parameter compared to sham, with an adjusted p-value to account for multiple comparisons using the Holm-Bonferroni method. We perform this test for all parameters, regardless of the outcome of the ANOVA, which can be found in the Results section.

2) The MEP analysis and results need to be reported with more details. The following points should be clarified in the revised version of the manuscript.2a) It is unclear how many MEPs (for paired-pulse, how many CS+TS and TS alone trials, respectively) were acquired per condition of each of the experiments. It reads like only 10 MEPs were acquired per condition (20 for paired-pulse blocks, so again 10 CS+TS and 10 TS alone). Given the known high variability of the MEP (cf. also subsection “Limitations” "the MEP variability we saw in some subjects") and the low number of participants (N = 12; which is understandable for TUS in humans but still low) these low MEP numbers are problematic. A larger number (20 or more MEPs) per condition would provide a much more stable estimate and allow the detection of small effects. Although there are quite some clinical neurophysiology papers out there with 10 or 15 MEPs, the kind of results presented here may shape the translation of TUS for human application and are thus too important for the community to suffer from low statistical power. Some of the findings (also of the negative ones) are in contrast to a previous study (Legon et al., 2018), and it is unclear whether this has to be attributed to differences in experimental design or simply noise. Many separate measures were obtained for the same subjects, and the effort of the authors is acknowledged, but maybe less measures and more trials would have been the better choice? Please report to point 1.

We thank the reviewers for these comments. As detailed in our response to point 1, we recruited an additional 4 new subjects, increasing the sample size to 16 for the whole study. We prioritized their experimental visits on acquiring an anatomical MRI (Visit 1) and testing the five key sonication parameters on single-pulse TMS experiments (visit 2). As suggested by the reviewers, we increased the number of trials per parameter (15-20 trials), as well as performing a supplementary behavioural control experiment (See point 8b). Unfortunately, given the current global halting of research activity for the foreseeable future, we were unable to ask participants to return for a third visit to perform paired-pulse experiments. We also note that the International Federation of Clinical Neurophysiology (IFCN) recommendation is to use 8 to 10 trials per condition for paired pulse TMS studies (Rossini et al., (2015)).

We have clarified the paired-pulse methods to clarify the number of trials per condition, total number of trials, and total sonication time per session

Materials and methods

“For each distinct paired-pulse paradigm (ie: SICI), the number of trials per participant were as follows: TS alone: 20 trials (10 TUS, 10 sham), CS-TS: 20 trials (10 TUS, 10 Sham). Each condition block consisted of 20 trials delivered in random order, and each participant received 8 paired-pulse blocks, with a cumulative sonication time of 40 seconds.”

Regarding the differences between our paired-pulse findings and other reports, we detail possible reasons in our Discussion section, which we attribute to two key differences in experimental methodology:

Discussion:

“Several methodological differences from our study should be noted, such as a time-locking the ultrasound to begin 100 ms prior to the conditioning stimulus, whereas we applied ultrasound for 490 ms prior to the first TMS pulse. Since sonication duration is a key parameter of TMS-mediated EMG suppression according to our findings, the longer sonication time in our experiment might have produced greater GABAA-mediated inhibition. In addition, we randomized the TS-alone condition with the paired pulse conditions in our experiment, whereas (Legon et al., 2018b) compared to a baseline MEP conducted in a temporally discrete block, potentially introducing variability inherent in MEP fluctuations over time (Ellaway et al., 1998).”

2b) The authors used the procedure of outlier exclusion. Outlier removal is a controversial method for MEP analyses. MEP amplitudes are not normally distributed but rather follow a power law, and removal of extremes is therefore corrupting the data. Was the pre-activation controlled and excluded together with the outliers? How many MEPs were left for the statistical analysis after exclusion? How do the results change when all trials are kept but the median is used per condition (instead of the mean) or the mean of log-transformed MEP values? Given that robustness is key for these kind of results, transformations should be avoided (or outlier removal at least).

The number of trial outliers excluded was small and only encompassed severe outliers (±2 standard deviations); moreover, this was applied equally for all sham and US conditions. For pre-activation, the outlier exclusion was similarly applied. We agree with the reviewer that there is no consensus as to whether outliers should be excluded or not in TMS studies; our methods are consistent with other TMS papers in the literature. Please see: doi: 10.1038/s41598-018-30480-z, 10.1016/j.brs.2019.05.015, 10.1152/jn.00762.2012, among others.

In our original 12 subjects, in single-pulse parameter experiments, participants received 10 trials per condition; at most 1 trial per condition were excluded, leaving 9 trials for analysis. Out of a total of 180 trials/participant, at most 2 outliers were identified, representing 1% of total trials.

For paired-pulse experiments, subjects received 10 trials per condition, at most 2 trials per condition were infrequently excluded, leaving 8 for analysis. Out of 160 total trials per participant, at most 3 outliers were identified, representing 2% of the total.

3) Several concerns have been raised regarding M1 targeting and the use of a normalized brain and MNI coordinates. Specifically, in the newly collected data, M1 (or specific parts of it) should be properly targeted by using neuronavigation and individual head models with T1/T2 (instead of a normalized brain and MNI coordinates). Moreover, the following points should be addressed in the revised version of the manuscript.3a) Why was a normalized brain used? You have a neuronavigation system and each participants' MRI. Why MNI coordinates? Should you not distinguish based on each participants' anatomy and TMS response to determine location?

Thank you for allowing us to clarify our use of neuronavigation, and to correct the erroneous phrase “normalized brain MRI”. The phrase is now corrected to:

Materials and methods:

“We captured the position of the coil in stereotactic space by registering the subject’s individual T1 anatomical MRI with the position of the TMS coil, using a TMS tracker and infrared camera via Brainsight. “

The method of selecting scalp position should also have been better elaborated, which we have done in the revised manuscript. Normalization to MNI coordinates was only done post-hoc via Brainsight software, solely for the purpose of generating Figure 2A (probabilistic average of electric field over a single standardized brain).

As the reviewer suggests, we did indeed use each participant’s TMS response to localize the position of the transducer on the scalp, and consequently the position of the TMS coil. As we now describe in the revised Materials and methods:

Materials and methods:

“At the beginning of each experimental session, we determined the FDI motor hotspot on the scalp, defined as the location over which TMS evoked MEPs of highest peak-to-peak amplitude in the target muscle at a given suprathreshold stimulator intensity (Bashir et al., 2013). A marker was used to precisely trace outline of the transducer holder on the scalp, to ensure that the angle and position of the wand was captured.”

In addition, we add:

“The ultrasound transducer was rigidly fixed to the underside of the figure-8 coil, and held in the centre of the coil, between the two windings. Previous validation studies (Opitz et al., 2015) showed that the measured electromagnetic maxima of a figure-of-8 TMS coil is located between the two coil windings, justifying our central placement of the transducer holder.”

Please see the individual acoustic simulation for validation (Figure 2—figure supplement 1) – we acknowledge in the limitations that the predicted sonication beam did not always encompass the anatomical location of the M1 cell bodies, which may account for individual variability in response to FUS. Nevertheless, the acoustic simulations show that our beam path targeted the primary motor cortex, and underlying white matter, in the majority of participants.

3b) Why was a mark placed on the head when you have a neuronavigation system? Since BrainSight neuronavigation was used to identify the TUS transducer position on the scalp, why was it not used to ensure and maintain correct transducer placement throughout the many measurements and sessions? Given the small diameter of the sonication beam, tiny changes in tilt or position can have a massive effect on the actually stimulated part of cortex.

Given the long stimulation blocks (e.g. 90 randomized frames with ISI 5s in the sonication duration experiment), our priority was to ensure that the position of the FUS-TMS stimulator results in consistent MEP amplitudes. As such, after locating the FDI motor hotspot, we used a permanent marker to precisely trace outline of the transducer holder on the scalp, to ensure that the angle and position of the transducer/TMS wand was captured. We agree with the reviewer that tiny changes in tilt or position in the transducer can have a massive effect on the actually sonicated part of cortex. Indeed, we found that small changes in tilt/position of the TMS coil can also significantly alter the consistency of generated MEPs, and this is precisely what we wanted to avoid. As such, our priority was to keep the entire TMS-FUS wand in a consistent position on the scalp for all stimulation experiments, since the outcome of interest was MEP amplitude. The Brainsight was used as a confirmatory tool, as well as enabling us to conduct acoustic and electromagnetic simulations post-hoc (ie: Figure 2A-B).

3c) Figure 2B: Please also provide sagittal and axial views to allow a better judgement of the targeting of M1. Is it actually targeting M1 or maybe premotor cortex? Which part of the precentral gyrus is actually sonicated?

We have revised Figure 2, adding both an improved coronal acoustic simulation (2B), as well as the reviewer’s requested axial and sagittal views derived from the neuronavigation targeting software (2C). As can be seen from these views, the transducer is overlying the hand knob of the precentral gyrus, at the location of the primary motor cortex. Please also see Figure 2—figure supplement 1 for individual coronal acoustic simulations which depict the estimated portions of the precentral gyrus sonicated for every individual.

3d) According to Fox, 2006 and Geyer, 1996 the motor cortex of human is allocated in the sulcus and at best to a small extent at the crown. In Figure 2 B the white matter is targeted as well as in Figure 3A where the pyramidal neurons are allocated in the white matter. Thus, the first part of the following sentence might not be accurate: "Similarly, the individual simulations of ultrasound propagation for each participant confirmed acoustic targeting of a portion of M1, as well as underlying white matter tracts. (Figure 2B)." see also "Our finding may suggest that cortical interneurons in layers II/III which are well-encompassed within the acoustic focus (Figure 4)"

We now provide individual acoustic simulations (Figure 2—figure supplement 1), as well as axial, sagittal, and coronal views (Figure 2B-C) of the targeted sulcus of a representative participant. Given the limited fidelity of acoustic simulation software and other limitations such as the lack of CT-derived bone densiometry, these are at best approximations. Furthermore, the wide focal length of the elongated acoustic focus (22.95 mm) predisposes to sonication of not only most cortical layers (including II/III), but also underlying white matter. Nevertheless, as we state in the Discussion, it is reasonable to speculate that based on our SICI results and others:

Discussion:

“Indeed, emerging electrophysiology work in animal models is suggesting that sensitivity to low-intensity TUS is mediated not only by parameter selection, but also that excitatory and inhibitory neurons have different sensitivities to sonication (Yu et al., 2019).”

Further studies are needed to refine the mechanism of action of TUS-mediated cortical effects.

3e) Without a CT scan and only T1-weighetd images no really reliable simulations can be obtained for the acoustic waves. Figure 2B only shows one "representative" subject. Have simulations been performed for all subjects or was the transducer only placed on top of the TMS M1 hotspot for each subject without modelling the sonication beam individually? This assumes that the relevant motor neurons of M1 are actually directly beneath the coil center, which is not necessarily the case.

Despite the superiority of CT imaging in calculating skull density and morphology metrics relevant to acoustic simulation, we did not wish to expose our healthy participant volunteers to radiation. We instead acquired anatomical T1 MRIs, which allow us the dual purpose of enabling subject-specific neuronavigation, and provided us with the ability to segment the skin, skull, and brain to estimate the acoustic pressure field. We now cite a validation study (doi: 10.1109/58.764862) showing agreement between hydrophone-acquired and simulated pressure fields using MRI data. We used a similar method to studies simulating trans-skull ultrasound propagation using segmented T1 MRI data (doi: 10.1121/1.5126685), and describe this in the revised methods and newly added Appendix 1. We agree with the reviewer that by simulating the skull as a homogenous tissue, the simulations show an imperfect estimation of the acoustic focus, and have added this to our subsection “Limitations”:

“Although our acoustic simulations were based on individualized brain imaging and transducer positions were captured by neuronavigation, the simulations are limited to two-dimensions, and the tissues are treated as homogenous layers due to absence of CT-derived density data which limits the fidelity of the estimated focus”.

3f) Figure 3A: The location and orientation of corticospinal output neurons in M1 is incorrect and misleading. They are actually located in the anterior bank of the central sulcus and oriented tangentially to the scalp. This should be corrected.

We have corrected the orientation and location of corticospinal neurons in our revised Figure 1A.

4) The phrasing "… a portion of M1…" is disconcerting. Because the transducer was concentric with the intersection of the TMS coil, wherever you put the coil is where the US was. Is that accurate? How did you confirm this however in your models if you did not use individual MRI but rather normalized MRI. MNI coordinates are mentioned previously but not given anywhere in the manuscript. TUS is highly localized and using generalized MNI coordinates is not appropriate. Please provide an acoustic wave modeling of the sonication for each participant.

We now provide an acoustic wave modelling for each participant (Figure 2—figure supplement 1) based on the scalp location of the transducer captured by neuronavigation. As mentioned in point 3a, the phrasing “normalized MRI” was misplaced in the manuscript. We only used normalized coordinates to generate the electromagnetic current map in Figure 2A. Individualized locations were used for acoustic simulation, and based on the simulation, a portion of the M1 gray matter, as well as underlying white matter was targeted for each participant with the sonication beam.

5) There is no data on how/if the transducer affected the TMS pulse or vice versa. This needs to be either collected or cited from Legon et al. 2018 and differences in transducer materials/design should be factored in if there are any.

We now include in our revised manuscript a more detailed description of the transducer material properties, dimensions, and mounting within the 70-mm TMS coil in the form of a supplementary Figure (Figure 1—figure supplement 1). We also cite Legon et al., 2018’s validation studies, which used the same 70-mm TMS coil as ours, and a custom transducer with similar dimensions and non-ferromagnetic properties. Of note, our transducer is constructed to be MRI-safe, and contains a non-ferromagnetic (brass) housing and is designed with a radiofrequency shield.

We now add, in our subsection “Limitations”:

“Lastly, we did not explicitly characterize the effect of the TMS-induced electromagnetic field on the operation of the ultrasound transducer, nor the effects of the transducer housing on the coil’s induced electrical field. However, such a characterization was rigorously performed in a prior study (Legon et al., 2018b) using the same TMS coil and a similarly-sized custom non-ferromagnetic ultrasound transducer, and found no significant effects in either direction.”

6) Gel pads are notorious for trapping air bubbles between interfaces. This can be easily detected using imaging mode of your transducer. Was this checked for? Was any other coupling media used?

Although our custom TUS transducer does not feature the software or hardware to allow image reconstruction, we have performed additional experiments with a dedicated imaging ultrasound system (Hitachi Aloka ProSound Α 7) to validate the lack of air bubbles between two relevant interfaces, using our methodology.

(Newly added Appendix 2)

Panel A: Visible air bubble at the TUS transducer-gel interface, indicated by the red arrow (left), and corresponding refraction and shadowing artifact seen when an imaging transducer is placed over the air bubble (right).

Panel B: Manual smoothing of the gel pad is performed until any small bubbles are extruded, leaving a homogenous black interface (left), and confirmatory imaging with ultrasound probe at the gel pad surface shows no artefact at the TUS transducer-gel interface (right).

Panel C: Application of gel pad over the frontal bone with the imaging transducer applied over the gel, visualizing the scalp-pad interface (left). Imaging of the scalp-pad interface, showing underlying tissue layers with no visible artefacts (right).

We now clarify this methodology in the revised Materials and methods:

“Aqueous compressible gel pads (Aquaflex, Parker Laboratories, NJ, USA) were cut into 1.5 mm thick pads 40 mm in diameter and placed between the surface of the transducer and the subject’s scalp. Any visible air bubbles at the transducer-pad interface were manually extruded, and a small amount of ultrasound gel (Aquasonic 100, Parker Laboratories, NJ, USA) was applied at the scalp-pad interface (Appendix 2)”.

7) TMS was applied in order to measure the cortical excitability changes with MEP. The TMS pulses were locked to the end of FUS or sham stimulation and they had an interstimulus interval of 5 seconds. If there was no jitter for TMS pulses it means that rTMS at 0.2 Hz was applied simultaneously with FUS. Repetitive TMS applied at a very low frequency of 0.2 Hz has been shown to be effective in several studies (Urushihara, 2006; Hosono, 2008). For example, rTMS over PMC led to an increase in somatosensory evoked potentials. Could the possible effect of low frequency rTMS on cortical excitability when applied simultaneously with FUS be discussed?

Thank you for raising this important question. We do not believe low-frequency rTMS will affect motor evoked potentials (MEP), as supported by three papers we cited and describe below:

1) Chen et al., 1997 conducted 0.1 Hz rTMS for 1 hour (360 pulses) and did not find motor cortical excitability changes.

2) Furukawa et al., 2010 carried out 100 sessions of M1 TMS at the same frequency as our inter-stimulus interval (0.2 Hz), and showed that MEP amplitude were neither enhanced nor reduced.

3) Cincotta et al.,2003 found that MEP size is not changed after 0.3 Hz rTMS intervention to M1 (total 540 pulses).

Overall, the rTMS frequency in these studies ranged from 0.1-0.3 Hz, with up to 540 pulses, without significantly altering MEP size. In our study, we tested sham and real FUS interventions with 15-20 frames per condition, with TMS delivered at 0.2Hz, and showed suppression of MEP size under certain condition. The short duration of TMS trains alone should not affect the MEP amplitude.

The two papers mentioned by the reviewer were done by the same group and show that 0.2 Hz rTMS can enhance the SEP N30 ratio. However, the stimulation target (pre-motor sensory cortex) and readout (SEP) were different from those in our study, which was the M1 and MEPs, respectively

8a) A further concern is "This sound was triggered every time a FUS or sham condition was delivered to the transducer." This could mean that the effects reflect acoustic TMS pairing, see e.g. doi: 10.3389/fnhum.2014.00398, other papers are around as well. Can the results in Figure 5D be due to longer acoustic stimulation? I expect the sham condition to be performed with the shortest duration, however not sure? Can this explain the lack of an effect in Figure 5E and F?

Thank you for the opportunity to clarify our choice of auditory control, and we detail our rationale in the Discussion. We recognize that several studies show that pairing TMS with an audible stimulus, especially human speech, can lead to increases in cortical excitability. Since the piezoelectric element within our ultrasound transducer emits a slightly audible buzzing tone when activated, also reported in other human FUS studies (eg: Legon et al., 2018), it was precisely to control for acoustic-TMS coupling that we decided to implement an audible masking sound in all conditions of the experiment.

To elaborate, we ensured that the audible masking sound was played for the same duration of time (0.5 seconds) for every experimental condition, including the sham, regardless of the parameters, relative to the TMS pulse. Furthermore, our sound did not consist of speech – rather it was a low-volume continuous high-pitched tone. Studies that have examined the TMS motor response of the hand area to different auditory stimuli (doi: 10.1046/j.1460-9568.2003.02774.x, Figure 3 (bottom) ) have shown that concomitant noise/tonal sounds do not alter the magnitude of MEP, while language perception significantly *increases* motor excitability. In agreement, studies examining TMS motor response of the leg (doi: 10.1016/j.neuropsychologia.2008.05.015, Figure 3) and tongue (doi: 10.1046/j.0953-816x.2001.01874.x) show a motor facilitation with semantic speech stimuli, but not with audible tonal sounds or noise. In our study, we did not find facilitation, but rather decreased excitability with application of transcranial ultrasound to the hand area of M1, while controlling for audible tones in all delivered conditions including sham.

The study referenced by the reviewer (doi: 10.3389/fnhum.2014.00398) found that acoustic-TMS pairing elicits significantly larger MEPs relative to a control condition without associated auditory stimulus. In all conditions within our experiments, the TMS pulse was preceded by an auditory stimulus of duration 0.5s, and we therefore expect that any TMS-acoustic pairing would be controlled for.

Interestingly, while writing these revisions, we came across a relevant preprint which examined audio masked vs. unmasked delivery of FUS/sham conditions (doi:10.1101/2020.03.07.982033). Of particular interest are Figure 1C and Figure 2A, showing that an audio-masked FUS sonication and audio-masked sham evoke *identical* auditory ERP's, whereas FUS alone without a mask evokes a significantly different ERP, and moreover can also be reliably differentiated from sham by participants. We believe this preliminary report corroborates our similar audio masking strategy, and highlights the importance of controlling for auditory confounds in future human ultrasound neuromodulation studies.

8b) Also: "and the task was more complex; nevertheless, the sonication parameters and cortical location were similar, and we observed an effect size of about 100 ms, though with higher variability." It may simply be that the start of the sonification sound leads via a kind of pre-triggering to shortened responses. This is discussed by the authors in subsection “Behaviour”. The effect in Figure 8A appears to be implausibly high. Control experiments seem to be reasonable with very light somatosensoric or close to threshold acoustic stimulation. The whole field of TMS-EEG suffers from acoustic and somatosensory contamination.

Although we control for auditory contamination with a masking sound, we agree with the reviewer that the start of sonication may contribute to a somatosensory pre-triggering, predisposing to shortened responses. We discuss this in Subsection “Behaviour”:

“Although the slightly audible sound of the piezoelectric transducer element was masked until participants could not distinguish between sham and active condition, we cannot rule out intersensory facilitation (Diederich and Colonius, 1987; Forster et al., 2002) as contributing to the difference in reaction times between conditions. Given that acoustic waves are mechanical in nature, it is conceivable that a subtle tactile sensation on the scalp at the site of transducer placement might act as an extra sensory cue which is difficult to replicate with a sham protocol that delivers no acoustic energy”

As suggested by the reviewer, we have conducted an additional control experiment with the additional 4 participants we recruited during the revision process. The results are seen in revised Figure 8, top panels. We analyze these results with paired t-test in the Results section, and discuss in subsection “Behaviour”:

**“**In a follow-up control experiment with 4 participants, the reaction times were not significantly different when a lower near-somatosensory threshold intensity (0.54 W/cm^2^) was used when compared to the 2.32 W/cm^2^ intensity used in all other experiments. Since individual somatosensory thresholds differ, and the threshold of transcranial acoustic energy necessary to affect motor tasks is unknown, further experiments are necessary to disentangle the role of somatosensory confounding in this behavioral task.”

In light of this control experiment, we have also added to the Abstract: “…and decreased reaction time on a visuomotor task compared to sham but not compared to a near-threshold intensity.”

[Editors' note: further revisions were suggested prior to acceptance, as described below.]

Summary:Fomenko et al., have thoroughly revised their manuscript. They added four new participants (now N = 16) with more trials (15-20 instead of 10 per condition), as well as more TUS modelling information. While this is not a large increase in sample size, the authors' efforts are appreciated. However, both the reviewers and the editors were concerned by the fact that the effects are robust for the 'block' experiment but not for the 'parameter' (or interleaved) experiment. Consequently, additional data should be collected in a sufficiently large sample (eg. keeping the N = 16 for interleaved and adding N = 16 for a blocked variation of stimulation parameters), targeting specifically the block vs. interleaved question for the duty cycle and pulse repetition frequency parameters.

Thank you for the opportunity to revise our manuscript. The reviewer and editor comments were insightful and have resulted in a stronger publication.

As requested by the reviewers, we have called back 16 healthy participants, in order to address the concern in the original manuscript; namely, that varying some sonication parameters (ie: PRF and DC) in blocked versus interleaved design yielded different effects on TMS-elicited MEP, whereas other parameters (ie: Sonication Duration) yielded robust effects regardless of experimental design.

For each participant, six new experiments were performed. The parameters examined were Duty Cycle (Sham, 10%, 30%, 50%), PRF constant duty cycle (Sham, 200Hz, 500Hz, 1000Hz), and PRF adjusted duty cycle (Sham, 200Hz, 500Hz, 1000Hz). For each individual parameter including sham, 15 replicate MEPs were acquired either serially in blocked fashion, or randomly interleaved with other parameters. We describe the protocol in detail in the Materials and methods section, and have used a rest period of 20 seconds between blocks, consistent with doi:10.1016/j.ultrasmedbio.2015.10.001. All parameters including sham were identically masked with a speaker tone lasting 0.5 during sonication. The results are discussed below in response to point 1.

Essential revisions:1) Regarding the block vs. interleaved question:If it makes indeed a difference for single trial MEP modulation whether TUS is applied in a block design with fixed parameters or with trial-by-trial variation of parameters, this would have important implications for TUS application and thus needs additional data collection, as well as detailed discussion. Specifically, for some parameters (sonication duration) the default parameter (0.5s) was effective in the interleaved design. However, for others (duty cycle) it was slightly different (10% instead of 30%), and for yet others (pulse repetition frequency) it was not different from sham anymore at all (but was previously with 1000 Hz). The blocked vs. interleaved interpretation is post-hoc and potentially valid only for duty cycle and pulse repetition frequency.

Please see additional figure added to the manuscript (Figure 5—figure supplement 1), as well as new results and in the Discussion section, where we write:

“Interestingly, our initial results show that although a particular combination of parameters can robustly suppress TMS-elicited cortical activity when applied in a sham-controlled block design (Figure 4), some parameters such as PRF when randomized and varied individually do not have the same robust effect, whereas others such as sonication duration do (Figure 5C-D). To address this discrepancy, we conducted follow-up experiments (Figure 5—figure supplement 1) in which we delivered three distinct parameter sets in blocked, and interleaved fashion. We found that sonication at 10% DC consistently results in suppression regardless of experimental design, whereas 30% DC only results in suppression when applied in blocked fashion. A duty cycle of 50% did not show any difference compared to sham regardless of experimental design. In contrast, we observed that sonication at three different pulse repetition frequencies (200, 500, and 1000 Hz) applied in a blocked design resulted in reduced cortical excitability compared to sham, whereas interleaving these parameters yielded no significant difference. These results held whether duty cycle was fixed, or whether the DC was adjusted to maintain an equal burst duration.

In summary, our findings suggest that when delivered in blocked fashion, longer sonication durations, lower duty cycles, and all three PRFs tested yield effective suppression of TMS-elicited MEPs. The trend towards greater suppression with increasing blocked PRF agree with recent literature, where higher PRF (1500Hz) in combination with low duty cycles were found to be more effective than lower PRF (300Hz) in neuromodulation of mouse motor cortex in vivo (King et al., 2013) and in vitro (Manuel et al., 2020). Furthermore, recent large animal studies reveal a bidirectional neuromodulation effects of varying TUS parameters (Yoon et al., 2019). As such, interleaving different parameters in short succession may lead to spillover effects, due to the random order of parameter delivery. From animal studies where randomized delivery of TUS parameters was studied, nonlinear effects were also found, which may be related to non-linear piezoelectric accumulation across the neural membrane capacity under the Neuronal Bilayer Sonophore model (Kim et al., 2014; Plaksin et al., 2014). In addition, excitatory or inhibitory changes in short-term plasticity may occur with repeated TUS stimulation, similar to those observed with repeated magnetic (Watanabe et al., 2014) and electrical stimulation (Udupa et al., 2016). These are currently under study in our laboratory and in emerging reports on LITUS short-term plasticity in animal models (Yu et al., 2019).”

Further experiments are ongoing in our lab to investigate the potential longer-term plasticity effects of FUS on the motor cortex, as well as potential nonlinear effects of individual FUS parameters.

2) The authors added 4 subjects to the existing data set (N = 12) but the reviewers fail to see the significance of this. First, data from the original 12 is presented in Figure 4 and the data from the new N = 4 is presented on its own. Could the authors explain the added value of this? It looks from the manuscript that N = 16 was inclusive for the parameter testing but not the 'basic parameter' testing. The authors could either remove Figure 4C or include that data in Figure 4A.

We presented the additional n=4 subjects in a separate panel of Figure 4 because these subjects received a different form of sham (inactive sham, or unpowered transducer facing the scalp) from the original n=12 subjects (active sham, or transducer flipped away from the scalp and powered, as in Legon, 2018). Given the active debate over the role of auditory confounds in the FUS neuromodulation literature as well as the first set of reviewer questions regarding our sham approach, we demonstrated that both forms of sham were able to suppress TMS-elicited MEPs, and then proceeded to use the Inactive sham for the remainder of the experiments, since it was less cumbersome and did not require manually flipping the transducer. We clarify in the revised Figure 4 caption:

“Effect of baseline ultrasound versus active and inactive sham on TMS-induced resting MEP amplitudes as measured by FDI EMG. A) Baseline parameters suppressed mean MEP amplitude compared to active sham, or powered transducer pointing upwards (p=0.002, paired t-test) N=12. B) Individual MEP values by participant by condition C) Baseline parameters suppressed mean MEP amplitude compared to inactive sham, or unpowered transducer pointing towards the scalp (p=0.012, paired t-test) N=4."

The reviewers are also directed to Figure 9, where the two types of sham are illustrated for clarity.

3) It is still unclear how many MEPs were left for each participant for the statistical analysis after exclusion. It looks like the new data uses 15 trials but the old 10 and the block experiment 20. Furthermore, it is stated in the Materials and methods that MEPs {plus minus} 2SD of mean were excluded. Please include M+/-SD in the Materials and methods.

The reviewers are correct in that new data uses either 15 or 20 trials per parameter, compared to the old data using 10. This was done as per the first revision reviewer request to increase the number of replicate trials per condition, and decrease the number of comparison.

In light of the reviewer comments regarding outlier exclusion, we have also chosen to now include all data, and use the median as the measure of central tendency for replicate trials in the TMS experiments including the new blocked vs. interleaved trials. Please see the Results section for our reanalysis, which now includes all data.

4) The following questions still need to be addressed:- Was the pre-activation controlled and excluded together with the outliers?- How do the results change when all trials are kept but the median is used per condition (instead of the mean) or the mean of log-transformed MEP values? Also, do the results hold when not removing the outliers?

As mentioned in Point 3, we have now decided to include all data and use the median as measure of central tendency; therefore, outliers are now no longer excluded. Please see revised manuscript and figures, where we demonstrate that our results hold.

- Why was the neuronavigation system not used to online maintain coil/transducer position? Coil angle cannot be reliably inferred from a felt pen drawing on the scalp.

We chose not to use online neuronavigation, which we now explicitly mention and discuss in the subsection “Limitations”. Our rationale was that we prioritized positioning the coil/transducer assembly at the TMS motor “hotspot”, to best achieve consistent motor-evoked potentials. We show in Author response image 1 the relevant markings we use on the TMS-TUS stimulator to ensure a precise surface positioning (contour tracing) and angulation (directional marking). Regarding the lateral angulation over the scalp, we maintain a perpendicular position to achieve maximal transducer coupling with the scalp.

5) Could the authors show the sonication beam model for all three slices (coronal, sagittal, axial)? The axial slices do not seem to be the most helpful ones when determining whether M1 was hit and which portion of it.

We now show the sonication beam model for the coronal and sagittal sections in the revised Figure 2B and 2D, showing a representative participant. Individual simulations for each participant coronal sections can be found in Figure 2—figure supplement 1.

Given that our study goal was not computational, and that the beam models are at best a two-dimensional approximation, which we mention in the limitations, we did not choose to pursue sophisticated 3D beam modelling, which we reserve for a future study, and therefore do not include an axial slice model. We also agree with the reviewers that the axial slices are not very helpful with respect to ascertaining which portion of the M1 was targeted. Instead, our 2D simulations serve as a post-hoc validation that our beam paths likely traversed the primary motor cortex and underlying corticospinal tracts.

[Editors' note: further revisions were suggested prior to acceptance, as described below.]

The new results demonstrate that the effectiveness of TUS parameters does indeed depend on whether TUS parameters are varied across trials or kept constant within a block. These additional findings are important and will be very useful for others using this approach. These results though also point to one important question that we believe should be in the published study: Are the results in the block design condition due to cumulative effects? That is, is there a systematic change over time, indicative of a cumulative effect? (To quote the reviewer who noted this concern, "Specifically, is there a build-up of suppression across trials within a block? This could be tested e.g. by regressing the MEP amplitude based on within-block trial number or by comparing early and late MEPs within a block. It would be a very important outcome to know whether the observed suppression is instantaneous or accumulating across trials.").

Thank you for pointing us toward an interesting analysis which might shed light into the potential mechanisms involved in our findings. In our revised manuscript, we now perform a post-hoc analysis of our blocked experiments, stratifying by trial number across all participants (Please see new Figure 5—figure supplement 2).

Similar to Volpert-Esmond et al., (2017), and also suggested by the reviewer, we then compare early trials (defined as the first 3 trials, or the first 20%) to late trials (last 3 trials, or last 20% of trials) of each parameter. We add the analytic methods and discussion to the revised manuscript as tracked changes.

On paired T-tests corrected for multiple comparisons, we do not detect a significant difference between the early and late trials across the three parameter sets. Please see the revised Results section for corrected p-values, which are non-significant.

We now add to our Discussion:

“To test whether a progressive accumulation of inhibition might be responsible for the observed difference in results between the blocked and interleaved study designs, we performed a post-hoc analysis of our blocked experiments, stratifying by trial number across all participants (Figure 5 – figure supplement 2). Within each parameter condition, we did not detect a significant difference between the magnitude of early and late trials and conclude that there is no temporal accumulation of MEP suppressive effects over blocks of fifteen trials, corresponding to a block length of approximately 90 seconds. Instead, the suppression appears to be instantaneous, with effective parameters increasing the likelihood of generating a lower TMS-elicited MEP, but not potentiating the effect of a subsequent stimulation. Notably, our blocked design features a 20 second pause between each set of 15 replicate stimulations – whereas interleaved delivery of random parameters involves an uninterrupted session of 60 random stimulations. We speculate that this 20 second rest period may lead to a resetting of cortical excitability to a more TUS-sensitive state. The lack of resetting could play a role in our observation of less robust effects when certain parameters are randomized in succession.”

Dedicated study designs will be needed to answer the question of both short-term and long-term effects of TUS, as well as its on-line and off-line interaction with TMS. Further questions to be answered include the role of somatosensory confounds/masking, as well as the use of on-line TMS navigation for targeting verification from trial to trial.